



# Case study of wave breaking with high-resolution turbulence measurements with LITOS and WRF simulations

Andreas Schneider[1], Johannes Wagner[2], Jens Faber[1], Michael Gerding[1], and Franz-Josef Lübken[1]

[1]Leibniz Institute of Atmospheric Physics at the University of Rostock (IAP), Kühlungsborn, Germany
[2]German Aerospace Center (DLR), Institute of Atmospheric Physics (IPA), Wessling, Germany

*Correspondence to:* Andreas Schneider (schneider@iap-kborn.de)

**Abstract.** Measurements of turbulent energy dissipation rates obtained from wind fluctuations observed with the balloon-borne instrument LITOS (Leibniz-Institute Turbulence Observations in the Stratosphere) are combined with simulations with the Weather Research and Forecasting (WRF) model to study the breakdown of waves into turbulence. Four flights from Kiruna (68° N, 21° E) and from Kühlungsborn (54° N, 12° E) are analysed. Average dissipation rates are in the order of $1\,\mathrm{mW\,kg^{-1}}$ ($\sim$0.1 K d$^{-1}$) with typically higher rates in the stratosphere compared to the troposphere. During two flights energy dissipation rates strongly decreased above the tropopause. One of these cases featured a patch with highly increased dissipation directly below the tropopause collocated with shear generation and wave filtering conditions. The second case showed nearly no turbulence at all above 15 km. For the other two flights, dissipation rates increased continuously across the whole ascent. For all flights, observed energy dissipation rates are related to wave patterns visible in the modelled vertical winds. Particularly, the drop in turbulent fraction for two of the flights mentioned above coincides with a drop in amplitude in the wave patterns visible in WRF. For other flights both dissipation rates and wave amplitudes show continuous distributions with height. This indicates small-scale partial wave breaking.

## 1 Introduction

Gravity waves transport energy and momentum and are thus an important factor in the atmospheric energetics. Typically, they are excited in the troposphere and propagate upwards and horizontally. Due to decreasing density, the amplitudes increase with altitude. Eventually, the waves break, producing turbulence and dissipation, and thereby depose their energy and momentum. This typically happens in the mesosphere. However, some waves already break in the stratosphere in spite of the stable stratification, e. g., at local instabilities due to wind shear. This modifies the energy flux from the troposphere to the mesosphere. A breaking wave is not always completely annihilated, but it may lose amplitude by transferring energy to smaller scales and eventually turbulence in a highly non-linear process (e. g. Franke and Collins, 2003).

The amount of energy deposited in the stratosphere by turbulent dissipation is largely unknown. A main reason is technical difficulties for measurements. The length scale where most dissipation occurs, also called the inner scale of turbulence $l_0$, is on the order of centimetres (or smaller) at stratospheric heights (e. g. Theuerkauf et al., 2011). This makes the direct observation technically challenging. For that reason, studies of stratospheric turbulence are sparse.



Measurements are performed in the lower stratosphere with radars (see Wilson, 2004, for an overview) and aircraft (e. g. Lilly et al., 1974; Hauf, 1993), in the lower and middle stratosphere with balloons (e. g. Barat, 1982; Theuerkauf et al., 2011), and in the middle and upper stratosphere with satellites (e. g. Gavrilov, 2013). In situ techniques have the advantage of much higher precision and resolution. To our knowledge, currently the only instrument for the direct in situ observation of turbulent

wind fluctuations in the middle stratosphere is the balloon-borne instrument Leibniz Institute Turbulence Observations in the Stratosphere (LITOS).

Wave breaking has been observed, e. g., in the lowermost stratosphere by Worthington (1998) and Pavelin et al. (2001) with radar and radiosonde. Plougonven et al. (2008) report mountain wave breaking over the Antarctic Peninsula. Franke and Collins (2003) observed gravity waves in the mesosphere with Na lidar and found upwards propagating waves still present

(with less amplitude) above an overturning region. Model studies of breaking gravity waves have, e. g., been carried out by Achatz (2005) and by Fritts and Wang (2013), Fritts et al. (2016), who performed direct numerical simulations (DNS) of a gravity wave superposed by fine-scale shear.

To study wave breaking into turbulence, a wide range of scales from kilometres (the wavelength of GWs) to millimetres (the viscous subrange of turbulence) has to be resolved. This cannot be performed by a single instrument. Thus several techniques

have to be combined. In this study, LITOS is used for the turbulence part and radiosonde observations from the same gondola for local atmospheric background conditions. To put the observations into a geophysical context and to obtain information about waves, regional model simulations with WRF (Weather Research and Forecasting model) driven by reanalysis data are applied. Four flights are analysed, thereof two from Kiruna (northern Sweden, 67.9° N, 21.1° E) and two from Kühlungsborn (northern Germany, 54.1° N, 11.8° E).

This paper is structured as follows: Section 2 gives an overview of the instrument LITOS and the data retrieval (Section 2.1) as well as the WRF model setup (Section 2.2). The results for four different flights are presented in Section 3. These are interrelated and discussed in Section 4, and finally conclusions are drawn in Section 5.

## 2   Instrumentation and model

### 2.1   Balloon-borne measurements

LITOS (Leibniz-Institute Turbulence Observations in the Stratosphere) is a balloon-borne instrument to observe small-scale fluctuations in the stratospheric wind field (Theuerkauf et al., 2011). The wind measurements are performed with a constant temperature anemometer (CTA) which has a precision of $cm\,s^{-1}$. It is sampled with 8 kHz yielding a sub-millimetre vertical resolution at 5 m ascent rate. Thus the inner scale of turbulence is typically covered. A standard meteorological radiosonde (Vaisala RS92 or RS41) is used to record atmospheric background parameters. LITOS was launched three times as part of

a ∼120 kg payload from Kiruna (67.9° N, 21.1° E) within Balloon Experiments for University Students (BEXUS) 6, 8 and 12 in 2008, 2009 and 2011, respectively (Theuerkauf et al., 2011; Haack et al., 2014; Schneider et al., 2015). The second generation of the small version of the instrument is an improvement of the one described by Theuerkauf et al. (2011) and consists of a spherical payload of ∼3 kg weight. It is suspended ∼180 m below a meteorological rubber balloon. Two CTA





sensors are mounted on booms sticking out at the top of the gondola. The instrument was launched several times from IAP's site at Kühlungsborn (54.1° N, 11.8° E), e. g. at 27 Mar 2014, 06 Jun 2014, and 12 Jul 2015.

In this paper, only flights are taken into account where data from more than one CTA sensor on the same gondola are available. Summarised, the data analysis is performed in three steps. First, the dissipation rate is retrieved similar as described by Theuerkauf et al. (2011). Then the $\varepsilon$ values from both sensors are compared to detect sections where one sensor is possibly affected by the wake of ropes. Finally, the remaining spectra are manually inspected to sort out cases were both sectors potentially have been in the wake.

The details of the retrieval are as follows: The data of the ascent is split into windows of 5 m altitude with 50 % overlap. In each window, the mean value is subtracted, and the periodogram is computed, which is an estimation of the power spectral density (PSD). The periodogram is smoothed with a Gaussian-weighted running average. The instrumental noise level is detected and subtracted. Initially, turbulence is assumed in each window and thus the Heisenberg (1948) model in the form given by Lübken and Hillert (1992) and Theuerkauf et al. (2011) is tried to fit to the observed spectrum (cf. Equation (A3) in Appendix A). If the fit succeeds, the inner scale $l_0$ is obtained. This leads to the energy dissipation rate $\varepsilon$ given by

$$\varepsilon = c_{l_0}^4 \frac{v^3}{l_0^4},\tag{1}$$

where $v$ is the kinematic viscosity (known from the radiosonde measurement) and $c_{l_0}$ is a constant depending on the type of sensor. The determination of $c_{l_0}$ for our sensor configurations is described Appendix A. Non-turbulent (or disturbed) spectra manifest in bad fits which are sorted out with the following set of criteria:

– The noise level detection fails, which usually means that the noise is not white, i. e. the periodogram is disturbed at small scales.

– $\varepsilon$ is negative; this may occur in very seldom cases when the spectrum is severely disturbed due to spurious effects.

– The mean logarithmic difference between data and fit exceeds a given threshold. This condition captures cases where the fit does not describe the data well, e. g. when no turbulence is present so that the periodogram does not follow form of the turbulence model.

– The inner scale $l_0$ lies outside the fit range. This means that the bend in the spectrum is not within the fit range and thus the fit is not meaningful. That can occur when the spectrum does not have the expected form of the turbulence model, when the inner scale lies at very small scales where the periodogram is dominated by noise, or when the periodogram is disturbed.

– The fit width is smaller than a threshold; in this case the fit is determined by too few data points.

– The value of the periodogram at $l_0$ is too close to the value of the noise level, i. e. too small a part of the viscous subrange is resolved.





- – The slope of the fit function at the small-scale end is less than a given threshold (less steep than $m^{-4}$, where $m$ is the vertical wave number). This indicates that the bend in the spectrum is not well covered by the fit and the data.

If one of the conditions applies, the spectrum does not follow the form for developed turbulence, thus $\varepsilon$ is set to zero. Due to the rigorous criteria the amount of detected turbulence can be considered a lower limit. Depending on the individual profile,

between 15 % and 62 % of the spectra of the single sensor profiles are classified as turbulent. Both sensors simultaneously yield turbulence for 12 % to 33 % of all data bins, depending on the flight. For the BEXUS flights much more turbulence is detected than for the flights with the small payload.

Sometimes a sensor has been in the wake of a rope supporting the gondola and the other sensor not, causing the $\varepsilon$ values of both sensors to differ by up to 5 orders of magnitude. To sort out such sections, altitude bins where the dissipation rate from

both sensors deviates by more than a factor of 15 are discarded, which amounts to roughly 8 % to 39 % of the valid spectra depending on the individual flight.

For the flights with the small payload, the remaining spectra have been inspected manually for sections where both sensors have been affected by the wake, and those that look suspicious have been taken out. A spectrum is regarded as wake-affected if it has a plateau in PSD near 10 cm spatial scale, which is estimated to be the extent of a Kármán vortex street originating from

the lines supporting the gondola. In this step, 62 of 1433 (113 of 975) spectra have been manually discarded for the flights from 27 Mar 2014 (12 Jul 2015), mainly in the troposphere and not above ∼20 km. This problem does not occur for the BEXUS flights, where the sensors were placed further away from the supporting lines. For all other altitude bins the average of both sensors is taken.

To quantify the stability of the atmosphere, the gradient Richardson number $Ri = N^2/S^2$ is used, which is the ratio of

the squared Brunt-Väisälä frequency $N^2$ and the square of the vertical shear of the horizontal wind $S^2$. The Brunt-Väisälä frequency can be written as $N^2 = \frac{g}{\Theta}\frac{d\Theta}{dz}$, where $\Theta$ is the potential temperature and $g$ the acceleration due to gravity. The wind shear is defined as $S^2 = \left(\frac{du}{dz}\right)^2 + \left(\frac{dv}{dz}\right)^2$, where $u$ and $v$ are the zonal and meridional wind components, respectively. The Richardson number represents the ratio of buoyancy forces (which suppress turbulence) to shear forces (which generate turbulence). According to a theory for plane-parallel flow established by Miles (1961) and Howard (1961), turbulence occurs

below a critical Richardson number of $Ri_c = 1/4$. The general applicability of that criterion was recently questioned based on measurements (e. g. Haack et al., 2014) and model simulations (e. g. Achatz, 2005). Often the shear is not strictly horizontal so that the theory by Miles (1961) and Howard (1961) is not applicable, as pointed out by Achatz (2005). However, it is still useful as an estimation of stability. In this study $Ri$ is retrieved from the radiosonde measurements. In order not to dominate the derivatives by instrumental noise, the potential temperatures and winds are smoothed with a Hann-weighted running average

over 150 m prior to differentiation with central finite differences.

## 2.2  Model simulations

Mesoscale numerical simulations are performed with the Weather Research and Forecasting (WRF) model, version 3.7 (Skamarock et al., 2008). Two nested domains with horizontal resolutions of 6 km and 2 km and time step 15 s and 5 s, respectively,





are applied. In the vertical direction 138 terrain following levels with stretched level distances of 80 m near the surface and 300 m in the stratosphere are used and the model top is set to 2 hPa (about 40 km altitude) for the BEXUS flights and 5 hPa (about 32 km altitude) for the flights from Kühlungsborn. At the model top a 7 km thick Rayleigh damping layer is applied to prevent wave reflections (Klemp et al., 2008), i. e. the top of the damping layer is the model top. Physical parametrisations

contain the Rapid Radiative Transfer Model longwave scheme (Mlawer et al., 1997), the Goddard shortwave scheme (Chou and Suarez, 1994), the Mellor-Yamada-Nakanishi-Niino boundary layer scheme (Nakanishi and Niino, 2009), the Noah land surface model (Chen and Dudhia, 2001), the WRF single-moment 6-class microphysics scheme (WSM6; Hong and Lim, 2006) and the Kain-Fritsch cumulus parametrisation scheme (Kain and Fritsch, 1990). The initial and boundary conditions are supplied by ECMWF (European Centre for Medium-Range Weather Forecasts) operational analyses on 137 model levels with

a temporal resolution of 6 hours. In WRF a temporal output interval of 1 hour is used. The computation of turbulent kinetic energy (TKE) is based on a prognostic equation which is solved additionally to the equations of motion and which includes transport, shear production, buoyancy production and dissipation terms. Shear and buoyancy terms include deformation and stability effects of the resolved flow and are related to turbulent motions by the horizontal and vertical eddy viscosities. The equation operates on the scale of the grid size. WRF Simulations are initialised 5 to 6 hours before the launch time of the

balloon.

## 3   Results

### 3.1   The BEXUS 12 flight (27 September 2011)

The BEXUS 12 flight was launched from Kiruna at 27 Sep 2011, 17:36 UT. The two left panels of Figure 1 show atmospheric conditions as observed by the radiosonde on board the payload. Temperatures decreased up to the tropopause at 10.3 km, ex-

cepting some small inversion layers. Above there was a sharp increase in temperature known as tropopause inversion layer (TIL) (Birner et al., 2002; Birner, 2006). Higher up, temperatures slightly decreased. Winds came from north-west near the surface and reversed between ∼6 km and 10 km. The reversal caused nearly opposite wind direction at 9 km altitude compared to 5 km, and a change of sign in both wind components. It further entailed strong wind shear below the tropopause, causing low Richardson numbers. Above the tropopause the wind field showed signatures of gravity wave activity with short wave-

lengths and no obvious altitude-dependent structure. In the stratosphere, Richardson numbers were generally larger than in the troposphere.

The right panel of Figure 1 depicts observed dissipation rates. Each blue cross corresponds to an altitude bin classified as turbulent (as described in Section 2.1). Overall, ∼30 % of the atmosphere was turbulent. The orange curve depicts a Hann-weighted running average over 500 m. Please note that especially in the stratosphere there are various bins with $\varepsilon = 0$ which

contribute to the running average but do not show up in the scatter plot. Dissipation rates varied over several orders of magnitude within only small altitude ranges (typically a few 10 m). This represents the well-known intermittency of turbulence. Mean dissipation rates were 2.7 mW kg$^{-1}$ in the troposphere and 3.5 mW kg$^{-1}$ in the stratosphere (excluding 1 km above and below the tropopause). Between 9 km and 10 km there was a thick layer with enhanced dissipation. As described above, this altitude





region featured low Richardson numbers caused by high wind shears. Thus turbulence was presumably induced by dynamic instability. Additionally, at this altitude a wind reversal was observed which caused filtering of gravity waves with respective phase velocities (if present). On the large scale, dissipation rates evinced an overall tendency to rise with altitude (cf. orange curve), excepting a step to smaller rates at the tropopause. For this step, two superposing causes are visible: (1) enhanced

stability in the TIL, and (2) the potential gravity wave filtering indicated by the wind shear below the tropopause mentioned earlier, which means that above less waves persist that can break and produce turbulence. On the other side, the wind shear is also expected to have generated new gravity waves, but these are expected to have small amplitudes.

Particularly in the stratosphere, turbulence occurred also for high Richardson numbers, in contradiction to the theory that $Ri \leq Ri_c = 1/4$ is necessary for turbulence. This behaviour is consistent with observations by Haack et al. (2014). In simulations

of gravity waves, Achatz (2005) found instabilities and onset of turbulence for Richardson numbers both smaller and larger than 1/4. He noted that the theory by Miles (1961) and Howard (1961) is not applicable to his simulations because the gravity wave phase propagation and thus the wave-induced shear is slanted. In the light of this comment, and taking into account that in the real atmosphere waves usually propagate inclined (i. e. the shear is not orthogonal to the altitude axis), the violation of the Richardson criterion for the LITOS measurements is comprehensible.

Figure 2 depicts results from WRF model simulations performed for the time and place of the flight (more precisely, snapshots at the middle of the ascent are shown). The upper left panel depicts horizontal winds at 850 hPa. Westerly winds flowed over the Scandinavian mountains which are expected to have excited mountain waves. Another potential source of gravity waves is geostrophic adjustment. Bending stream lines are visible, e. g., over the Scandinavian mountains, west of the flight track. The upper right panel presents a vertical section of horizontal winds and potential temperatures. It visualises that the jet

(∼7 km to 10 km altitude) had a local structure and involved strong wind shears.

With a grid resolution of 2 km WRF can resolve waves with horizontal wavelengths larger than about 10 km. These waves can be seen, e. g., in the vertical winds, which are used as a proxy. This quantity is plotted in the lower left panel of Figure 2. Strong wave-like patterns are visible especially over the Scandinavian mountains, which correspond to the mountain wave excitation mentioned above. Weaker wave patterns are visible near the flight trajectory, downstream of the mountains. Between

roughly $x = 400$ km and $x = 550$ km, the wave patterns change at tropopause height (approximately 10 km altitude): Above there is less amplitude than below. This is ascribed to the wave breaking and filtering mentioned before. Further upwards the amplitude increases slowly.

Waves can propagate over considerable distances and times. Therefore it is not sufficient to look at potential sources in the vicinity of the flight track. Even if sources are found, the waves may have propagated to other places (away from the point of

interest), while waves from sources outside the domain may have propagated to the location of observation. For resolved waves the model takes care of these issues. Waves seen in WRF at the location of the flight may have travelled from remote places, yet the important information is not their origin, but that they were present during the measurement.

To trigger turbulence, wave *breaking* is necessary. Such events are triggered by dynamic or convective instabilities or by wave-wave interactions (e. g. Fritts and Alexander, 2003). In WRF, the break-down to turbulence is parametrised by solving a

prognostic equation for turbulent kinetic energy (TKE), which is based on production terms due to shear and buoyancy obtained





from the resolved flow. TKE is plotted in the lower right panel of Figure 2. It peaks near 10 km height at the location of the flight. This corresponds nicely to the intense turbulent layer observed by LITOS. It is reproduced in WRF due to the shear instability on scales resolved by the model. That highlights the geophysical significance of that layer. With LITOS, weaker turbulence is observed over the whole altitude range (i. e. below 10 km as well as above). This background turbulence is not covered by

the model, because it is caused by shear and buoyancy instabilities of the mean flow on scales smaller than resolved by the model. In the stratosphere, some layers are present with dissipation rates in similar order as observed near 10 km height, but these are relatively thin and are not associated with $Ri < 1/4$. For example, there is a layer with large dissipation rates between $\sim$22.48 km and 22.63 km altitude, but it is only $\sim$150 m thick, and Richardson numbers are around 1. In the stratosphere, the vertical model resolution is 300 m. Thus it is reasonable that the layer at 22.5 km is not reproduced in WRF with enhanced

TKE.

### 3.2 The BEXUS 8 flight (10 October 2009)

LITOS was previously flown on BEXUS 8, launched from Kiruna at 10 Oct 2009, 08:03 UT. Haack et al. (2014) already describe some features of that flight, mainly statistics about turbulent layers as well as dissipation rates and their relation to Richardson numbers. Please note that they computed dissipation profiles with a 25 m window, while here a 5 m window, an

updated value of the constant $c_{l_0}$ in (1), and an updated set of quality criteria is used. Here, the focus lies on the comparison with other flights and WRF simulations.

Figure 3 presents the observations. The temperature structure from the radiosonde data shows a tropopause at 8.1 km, i. e. considerably lower than for BEXUS 12, and only small local sections with increasing temperature above. Winds came from north western directions below $\sim$20 km and from south west above. No zonal wind reversal as for BEXUS 12 was present.

Energy dissipation rates are plotted in the right panel of Figure 3. Again $\varepsilon$ is intermittent. In contrast to BEXUS 12, no pronounced maximum in dissipation is visible. This is consistent with the absence of a wind reversal or large wind shear. Richardson numbers are variable; mostly values are much larger than the critical number 1/4 in the entire troposphere and stratosphere, only some small layers with $Ri < 1/4$ are present. There is no extended region with $Ri < 1/4$ as for BEXUS 12 near 10 km altitude. Average dissipation rates are 2.0 mW kg$^{-1}$ in the troposphere, and 5.5 mW kg$^{-1}$ in the stratosphere (not

taking into account the tropopause region 1 km above and below the tropopause).

Model simulations for the BEXUS 8 flight are presented in Figure 4. Tropospheric winds flowed against the Scandinavian mountains from western directions, but were weaker than during BEXUS 12. No jet was present. The expected mountain waves are visible in the vertical winds. In the lee of the mountains, wave patterns with smaller amplitudes are present at the location of the flight track. They intensify above altitudes of $\sim$20 km. No drop in wave amplitude similar to that during BEXUS 12 at

$\sim$10 km is visible. This is consistent with no wave filtering and moderate dissipation rates throughout all altitudes with no peak in dissipation during BEXUS 8. The model TKE shows no enhancement outside the boundary layer, consistent with no wave filtering and no pronounced maximum in dissipation.



### 3.3 The 27 March 2014 flight

A small LITOS payload of second generation was launched from Kühlungsborn at 27 Mar 2014, 10:10 UT.

The left panel of Figure 5 shows temperatures smoothed over 15 data points ($\sim$150 m) as well as zonal and meridional winds. The smoothing is necessary because for this flight the temperature measurement is perturbed by radiation effects as

the radiosonde was incorporated in the main payload; these effects get worse with increasing altitude. Temperatures decreased up to the tropopause at 9 km. Between 9 km and $\sim$30 km altitude they stayed nearly constant and started to increase further upwards. Winds were easterly and turned northwards above $\sim$20 km altitude. A strong southeasterly jet was present between $\sim$6 km and 10 km height. Superposed are signatures of small-scale gravity waves. Wind shears originating from the jet may have excited turbulence and/or waves. The effect of the shear is visible as a layer with enhanced dissipation at this altitude

(see below). Richardson numbers are shown for altitudes below 9 km only because they involve derivatives of the temperature profile which was disturbed by radiation effects as described above.

Dissipation rates are presented in the right panel of Figure 5. The data below 650 m altitude are affected by the launch procedure (precisely the unwinding of the dereelers) and are thus discarded and not shown in the plot. $\varepsilon$ values show the well-known intermittency similar to the BEXUS flights. The running average shows some structure in the troposphere, e. g. a few

layers that are standing out with larger rates. Most prominently this can be seen near 8 km. That is in the same altitude as the wind shear due to the jet, which speaks for shear-induced turbulence. Precisely, there were two turbulent layers from 7.5 km to 7.9 km and from 8.1 km to 8.3 km height; within both, Richardson numbers were below 1 and partly below 1/4. Other sheets with large dissipation were, e. g., near 6.1 km and around 3.0 km altitude. In the lower stratosphere dissipation rates increased with altitude, while the variation was smaller compared to the troposphere. Mean values are 0.50 mW kg$^{-1}$ in the troposphere

and 4.0 mW kg$^{-1}$ in the stratosphere.

Figure 6 depicts results from WRF simulations for the time of the flight. The upper left panel shows horizontal winds at 850 hPa, which were easterly or south-easterly. In the upper right panel horizontal winds are depicted as altitude section, showing that the strong jet had not much structure in horizontal direction, while the sharp vertical structure is reproduced as observed by the radiosonde. The lower left panel shows a vertical profile of vertical winds. Wave patterns are visible, which

stretch over the whole altitude range. Particularly, a superposition of a wave with long vertical wavelength ($\lambda_z \approx 8$ km) and nearly horizontal phase fronts and waves with short horizontal wavelength (10 km to 20 km) and phase fronts in the vertical can be seen. The occurrence of wave patterns corresponds to medium energy dissipation observed throughout all altitudes. The lower right panel of Figure 6 shows the TKE. Outside the boundary layer there is an enhancement near 7.5 km altitude. It corresponds nicely to a thick, strong turbulent layer in the measurement by LITOS between $\sim$7 km and 8.5 km height. Within

this observed turbulent layer, which in fact consists of several layers, Richardson numbers are smaller than 1 almost everywhere and partly even smaller than 1/4.



### 3.4 The 11/12 July 2015 flight

A night-time flight with LITOS was performed at 11/12 Jul 2015 from Kühlungsborn, launched at midnight local time (22:01 UT at 11 Jul). The radiosonde was positioned 60 m below the main payload to avoid disturbances of the temperature sounding. The observed background parameters are depicted in the two left panels of Figure 7. Westerly winds prevailed up to ∼19 km altitude, whereas above winds came from the east. This change in direction was not associated with a significant wind shear because velocities were small in that altitude region. A jet is visible at about 10 km height. Superposed on the winds are signatures of small-scale gravity waves. Above the tropopause at 11.3 km altitude there was a small tropopause inversion layer. Higher up temperatures remained rather constant up to ∼20 km, where they started to increase.

Richardson numbers were typically lower than for the other flights, indicating less stability. There are several layers where the Richardson number is below the critical limit of $Ri_c$ (1/4). These layers are relatively thin.

Energy dissipation rates (data below 550 m are excluded due to disturbances from the launch procedure) showed a strong layer structure, with enhanced dissipation at, e. g., ∼2.0 km, 3.8 km, 7.2 km, 8.9 km, 11.0 km, 12.1 km, and 14.3 km. These layers of intense turbulence mostly corresponded to Richardson numbers smaller than $Ri_c = 1/4$, or at least to $Ri < 1$. Above ∼15 km altitude, hardly any turbulence was detected; only a few thin turbulent layers were observed. Thus above 15 km the average dissipation rate (for which no turbulence is counted as zero) was only 0.01 mW kg$^{-1}$, while below 15 km it was 0.64 mW kg$^{-1}$.

Results from corresponding WRF simulations are depicted in Figure 8. Horizontal winds at the 850 hPa level were mainly westerly. The altitude section shows that the strong jet did not have much variation in the horizontal direction. Vertical winds reveal wave patterns that are particularly intense around the tropopause and gradually become weaker near ∼15 km, with less amplitude above. This drop in wave amplitude is at the same altitude as the drop in observed dissipation. The TKE has enlarged values around 3 km altitude and near the tropopause, however the enhancement is small at the flight path. Correspondingly, the thickness of the strong turbulent layers detected by LITOS is relatively small; that means that these dissipative layers are potentially not resolved in the model.

## 4 Discussion

A comparison of the observed dissipation profiles and the wave patterns in the model vertical winds for the different flights yields that more turbulence observed by LITOS comes along with stronger wave patterns visible in WRF, and vice versa. Particularly, this can be seen at the BEXUS 12 flight (27 Sep 2011) at the jump in dissipation and wave amplitude at ∼10 km altitude. In this case, the involved mechanism is a shear instability and potential wave filtering shortly below. At 12 Jul 2015, average dissipation rates drop at ∼15 km, and so does the wave amplitude visible in WRF. A similar feature has been observed during another flight at 06 Jun 2014 (not shown): Likewise, LITOS data exhibit a sharp drop in turbulence at ∼15 km, and the corresponding WRF simulation shows strong wave patterns below ∼15 km and very weak ones above. In contrast, the flights from 10 Oct 2009 and 27 Mar 2014 do not show such a drop in dissipation rate or wave amplitude. For these two flights,





moderate dissipation rates as well as wave amplitudes continue throughout all altitudes, with a slight increase towards higher altitudes.

The relation between waves and turbulence can also be seen in averages. Table 1 summarises mean dissipation rates from LITOS and mean absolute vertical fluxes in WRF for the flights presented in Section 3. For 12 Jul 2015, average dissipation rates above 15 km are more than two orders of magnitude lower than for the other flights. Below 15 km, mean $\varepsilon$ values are in the same order of magnitude for all flights. At 12 Jul 2015 average dissipation rates below and above 15 km deviate by nearly two orders of magnitude. Consistently, the average absolute vertical flux above 15 km is lowest for all flights, and the values below and above 15 km deviate by one order of magnitude. At 27 Mar 2014 the fluxes below and above 15 km only deviate by a factor of 2.5. For the BEXUS flights (10 Oct 2009 and 27 Sep 2011), both dissipation rates and fluxes are on average larger than for the flights from Kühlungsborn (27 Mar 2014 and 12 Jul 2015).

We interpret this behaviour as continuous partial wave breaking, meaning that a wave continuously loses amplitude by transferring energy to smaller scales and eventually turbulence due to non-linear processes. Partial wave breaking has been observed by lidar and described by Franke and Collins (2003). They found regions of strong overturning, and upwards propagating waves present below as well as (with less amplitude) above the overturning region. They argue that, depending on the amplitude, a breaking wave is not always completely annihilated, but the amplitude may be modulated in a highly non-linear event. Nappo (2002, p. 125) states that "gravity wave and turbulence are often observed to exist simultaneously." Via the process of continuous wave breaking, the occurrence of waves is connected to the intensity of turbulence. Pavelin et al. (2001) observed intense turbulence in the lowermost stratosphere during a period of maximal wave intensity using radar at Aberystwyth (52.4° N, 4.0° W), which supports the above hypothesis.

Mean dissipation rates observed by LITOS are in the order of $10^{-3}\,\mathrm{W\,kg^{-1}}$ (roughly $0.1\,\mathrm{K\,d^{-1}}$). This is an order of magnitude below typical solar or chemical heating rates which are in the order of $1\,\mathrm{K\,d^{-1}}$ (Brasseur and Solomon, 1986, Fig. 4.19b). However, within thin layers rates of $10^{-1}\,\mathrm{W\,kg^{-1}}$ ($\sim 10\,\mathrm{K\,d^{-1}}$) are observed, which is larger than solar heating. The low mean energy dissipation rates are not explicitly contained even in high-resolution models, which cannot describe the large intermittency. Only large layers with highly increased dissipation as encountered during BEXUS 12 are captured.

Observed dissipation rates are partly larger than those reported by other publications using different methods. Barat (1982) obtained values between $1.4 \times 10^{-5}\,\mathrm{W\,kg^{-1}}$ and $3.9 \times 10^{-5}\,\mathrm{W\,kg^{-1}}$ from balloon measurements. Wilson et al. (2014) found $\varepsilon$ values between $3 \times 10^{-5}\,\mathrm{W\,kg^{-1}}$ and $6 \times 10^{-4}\,\mathrm{W\,kg^{-1}}$ in the upper troposphere from radar measurements. These are lower rates than the averages in this work, but within the range of the variability. Lilly et al. (1974) observed stratospheric dissipation rates between $7 \times 10^{-4}\,\mathrm{W\,kg^{-1}}$ and $2 \times 10^{-3}\,\mathrm{W\,kg^{-1}}$, depending on the underlying terrain, with aircraft. These results are in similar order of magnitude as the averages in this study. Haack et al. (2014) reported mean dissipation rates of $2 \times 10^{-2}\,\mathrm{W\,kg^{-1}}$ for the BEXUS 6 balloon flight and $5 \times 10^{-3}\,\mathrm{W\,kg^{-1}}$ for BEXUS 8 for the altitude range 7 km to 26.5 km, using a slightly different retrieval. That their average value for BEXUS 8 is similar to the one in this study is a consequence of two compensating effects: The new retrieval with more rigorous quality control criteria yields more spectra classified as non-turbulent which contribute to the average with $\varepsilon = 0$, yet the updated value of the constant $c_{l_0}$ in Equation (1) (cf. Appendix A) yields higher dissipation rates by a factor of $\sim 50$ for the same $l_0$.



## 5 Conclusions

In this paper high-resolution turbulence observations with LITOS are complemented by model simulations with WRF to study the relation between turbulence, waves, and background conditions. Four flights are selected where in each case data from two wind sensors are available; this allows a high quality assurance.

Enhanced energy dissipation rates were observed where pronounced instabilities were detected by the radiosonde. Moreover, measured shear instabilities and associated enhancements in dissipation on scales resolved by WRF also coincide with enlarged model turbulent kinetic energies (TKE). For instance, during the BEXUS 12 flight (27 Sep 2011), a wind reversal was observed which caused a large shear instability (indicated by Richardson numbers smaller than 1/4) as well as potential wave filtering. The resulting turbulence was detected by LITOS as a region with strongly enhanced dissipation rate. The model turbulent

kinetic energy (TKE) peaks in this region, highlighting the significance of that layer. When looking at the vertical winds from WRF, wave patterns change at that height with large amplitudes below and small ones above; this again suggests the occurrence of wave breaking. Thus in this case the geophysical cause of the observed turbulent layer is clearly visible. The large scale instability is resolved by the radiosonde and the model. On the other hand, many other (less intense) turbulent layers observed by LITOS are obviously too thin to be related to the much coarser data of the radiosonde or the WRF results.

A relation between turbulence detected by LITOS and the presence of wave-like structures in WRF is noted: For the available summer flights at 06 Jun 2014 (not shown) and 12 Jul 2015, hereafter scenario 1, a drop in turbulence occurrence at approximately 15 km altitude with hardly any turbulence above was observed. In contrast, no such feature was present at the other flights (scenario 2; 10 Oct 2009, 27 Sep 2011, and 27 Mar 2014), i. e. turbulence occurred at all altitudes. In the associated model simulations, wave signatures become weaker around 15 km for scenario 1 (06 Jun 2014 and 12 Jul 2015), while they

continue throughout all altitudes for scenario 2 (the other flights). Altogether, observed dissipation generally is weaker during lower wave activity (as seen in WRF), and larger where larger wave amplitudes are seen. These findings can be explained by a continuous fractional wave breaking.

    The above hypothesis is made based on the limited dataset from a few flights. More flights at selected meteorological situations are planned to further study such a connection. Moreover, a direct measurement of gravity wave activity in combination

to the turbulence observations is preferable.

## Appendix A: Derivation of the constant $c_{l_0}$ in Equation (1)

To retrieve energy dissipation rates from observed spectra, relation (1) between inner scale $l_0$ and dissipation rate $\varepsilon$, $\varepsilon = c_{l_0}^4 \nu^3 / l_0^4$, and especially the value of the constant $c_{l_0}$ is important. To obtain correct values, care has to be taken of which component(s) of the spectral tensor are observed. In the following, the derivation of the constant $c_{l_0}$ is summarised.

In the inertial subrange, the longitudinal component, transversal component, and trace of the structure function tensor for velocity fluctuations have the form

$$D_{xx}(r) = C_{xx}r^{2/3}, \tag{A1}$$





where $xx$ is a placeholder for rr (longitudinal), tt (transversal), or $ii$ (trace), and the structure constant has the form $C_{xx} = b_{xx} a_v^2 \varepsilon^{2/3}$ with $b_{rr} = 1$, $b_{tt} = \frac{4}{3}$, $b_{ii} = b_{rr} + 2b_{tt} = \frac{11}{3}$ (Tatarskii, 1971, p. 54ff) and the empirical constant $a_v^2 = 2.0$ (e. g. Pope, 2000, p. 193f). In the viscous subrange, the structure function is

$$D_{xx}(r) = \tilde{C}_{xx} r^2 \tag{A2}$$

with $\tilde{C}_{xx} = c_{xx} \frac{\varepsilon}{v}$ and the factors $c_{rr} = \frac{1}{15}$, $c_{tt} = \frac{2}{15}$, $c_{ii} = c_{rr} + 2c_{tt} = \frac{1}{3}$ (Tatarskii, 1971, p. 49).

Based on Heisenberg (1948, (28)), Lübken and Hillert (1992, (4)) gave a form of the temporal spectrum in the inertial and viscous subranges, which reads for velocity fluctuations

$$W(\omega) = \frac{\Gamma(\frac{5}{3}) \sin(\frac{\pi}{3})}{2\pi u_b} C_{xx} \frac{(\omega/u_b)^{-5/3}}{\left(1 + \left(\frac{\omega/u_b}{k_0}\right)^{8/3}\right)^2} \tag{A3}$$

where $u_b$ is the ascent velocity of the balloon and $k_0$ denotes the breakpoint between inertial and viscous subrange. The
normalisation is obtained by considering the limit $k \ll k_0$ for the inertial subrange. Using the relation $\Phi(k) = -\frac{u_b^2}{2\pi k} \frac{dW}{d\omega}(k u_b)$ between temporal and spatial spectrum (Tatarskii, 1971, (6.14)), the corresponding three-dimensional spectrum is

$$\Phi_{xx}(k) = \frac{1}{6\pi} \frac{\Gamma(\frac{5}{3}) \sin(\frac{\pi}{3})}{2\pi} C_{xx} k^{-11/3} \frac{5 + 21\left(\frac{k}{k_0}\right)^{8/3}}{\left(1 + \left(\frac{k}{k_0}\right)^{8/3}\right)^3}. \tag{A4}$$

The constant $c_{l_0}$ in (1) can be computed from the condition of the structure function at the origin

$$\frac{d^2 D_{xx}}{dr^2}(0) = \frac{8\pi}{3} \int_0^\infty \Phi_{xx}(k) k^4 \, dk \tag{A5}$$

(Tatarskii, 1971, p. 49f). Inserting the structure function (A2) and the spectrum (A4) into condition (A5), integrating and solving for $1/k_0$ yields

$$l_0 = \frac{2\pi}{k_0} = \underbrace{2\pi \left(\frac{3}{16} \Gamma(5/3) \sin(\pi/3) \frac{b_{xx}}{c_{xx}} a_v^2\right)^{3/4}}_{=c_{l_0}} \left(\frac{v^3}{\varepsilon}\right)^{1/4}. \tag{A6}$$

CTA wire probes are sensitive perpendicular to the wire axis but insensitive parallel to the wire axis. For the earlier flights, the wires of the CTA sensors were oriented vertically so that they are sensitive in both horizontal directions and insensitive in
the vertical direction, i. e. for an ascending balloon both transversal components are measured. Thus $b_{xx} = 4/3 + 4/3 = 8/3$ and $c_{xx} = 2/15 + 2/15 = 4/15$, which leads to $c_{l_0} = 14.1$. For the flight at 12 Jul 2015, one sensor with the wire oriented horizontally was flown, which is sensitive in the vertical and one horizontal direction yet insensitive in the other horizontal direction (parallel to the wire). In this case $b_{xx} = 1 + 4/3 = 7/3$ and $c_{xx} = 1/15 + 2/15 = 3/15$ so that $c_{l_0} = 15.8$.

Haack et al. (2014, Section 4) used different components of the structure function constant yielding $c_{l_0} = 5.7$. Since in (1)
the constant occurs with $c_{l_0}^4$, this results in a difference in $\varepsilon$ of a factor of $\sim 50$ for the same $l_0$.



*Acknowledgements.* The data of the BEXUS 8 flight were kindly provided by Anne Haack. The BEXUS programme was financed by the German Aerospace Center (DLR) and the Swedish National Space Board (SNSB). We are grateful for the support by the "International Leibniz Graduate School for Gravity Waves and Turbulence in the Atmosphere and Ocean" (ILWAO) funded by the Leibniz Association (WGL).



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



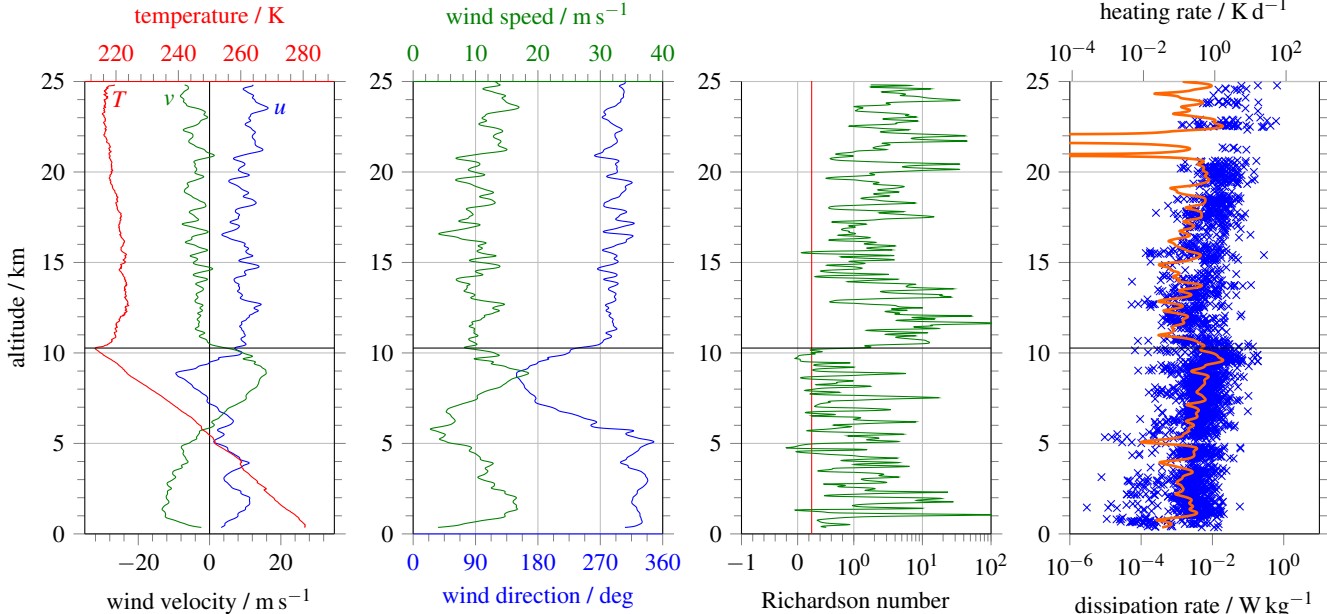

**Figure 1.** Observations during the BEXUS 12 flight. Left: Zonal winds $u$ (blue), meridional winds $v$ (green) and temperatures $T$ (red) from the radiosonde. Centre left: Wind direction (blue) and horizontal wind speed (green) from the radiosonde. Centre right: Richardson number $Ri$ computed from the radiosonde data, using a smoothing over 150 m prior to differentiation. The $Ri$ axis is split at 1 into a linear and a logarithmic part. The red line shows the critical Richardson number 1/4. Right: Energy dissipation rates $\varepsilon$ observed by LITOS. The blue crosses mark single turbulent spectra computed on a 5 m grid with 50 % overlap, the orange curve shows a Hann-weighted running average over 500 m (non-turbulent bins count as zero in the average). The horizontal black line in all four panels marks the tropopause.

**Table 1.** Average dissipation rates observed by LITOS and mean absolute values of vertical energy fluxes from the WRF model. The fluxes are taken from a $y$ section through the launch point averaged over the $x$ coordinate in an area 50 km east and west of the launch point and over altitude from 7.5 km to 12.5 km ($< 15$ km) or 17.5 km to 22.5 km ($> 15$ km).

| Flight | | mean dissipation rate / mW kg$^{-1}$ | | | | | mean vert. flux / W m$^{-2}$ | |
|---|---|---|---|---|---|---|---|---|
| Date | Place of launch | tropo | strato | all | $< 15$ km | $> 15$ km | $< 15$ km | $> 15$ km |
| 10 Oct 2009 | Kiruna | 2.0 | 5.5 | 4.4 | 2.1 | 6.9 | 0.18 | 0.073 |
| 27 Sep 2011 | Kiruna | 2.7 | 3.5 | 3.5 | 3.0 | 4.2 | 0.23 | 0.028 |
| 27 Mar 2014 | Kühlungsborn | 0.50 | 4.0 | 3.1 | 1.1 | 4.6 | 0.038 | 0.015 |
| 12 Jul 2015 | Kühlungsborn | 0.85 | 0.02 | 0.34 | 0.64 | 0.01 | 0.064 | 0.0069 |




**Figure 2.** Map of horizontal winds at 850 hPa (upper left), vertical section of horizontal winds (upper right), vertical section of vertical winds (lower left), and vertical section of turbulent kinetic energy (TKE) (lower right) from WRF simulations for 27 Sep 2011, 18:00 UT. The black curves visualise the trajectory of the BEXUS 12 flight. In the upper left panel, the blue streamlines show the wind direction, the white lines visualise coastlines and a latitude/longitude grid, and the black line indicates the location of the vertical sections. In the upper right panel, the white isolines show potential temperature.





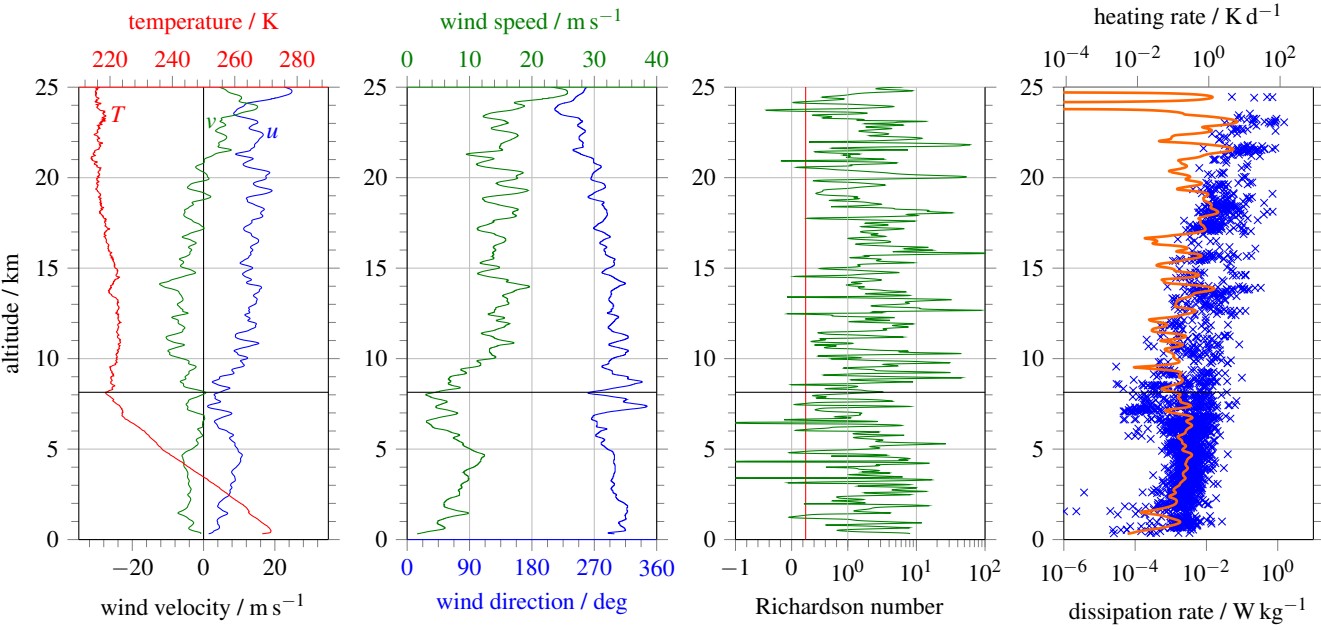

**Figure 3.** Same as Figure 1, but for the BEXUS 8 flight (10 Oct 2009)





**Figure 4.** Same as Figure 2, but for WRF simulations for 10 Oct 2009, 9:00 UT and showing the trajectory of the BEXUS 8 flight. Please note that for the TKE the colourbar is scaled differently than in Figure 2.

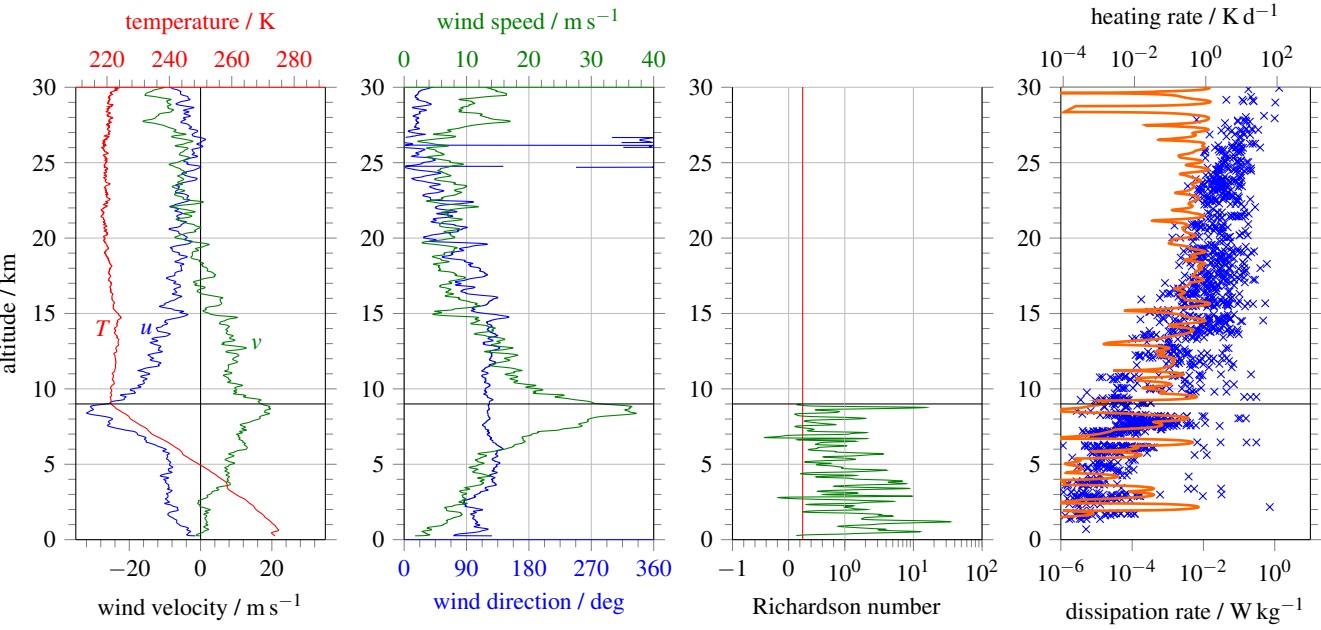

**Figure 5.** Same as Figure 1, but for the flight from Kühlungsborn at 27 Mar 2014. Due to disturbances of the temperature data, temperatures are smoothed in the plot in the left panel, and Richardson numbers are shown only for altitudes lower than 9 km. The dissipation profile excludes the lowermost 650 m due to disturbances from the launch procedure (dereeling of the payload suspension).



**Figure 6.** Same as Figure 2, but for WRF simulations for 27 Mar 2014, 11:00 UT.





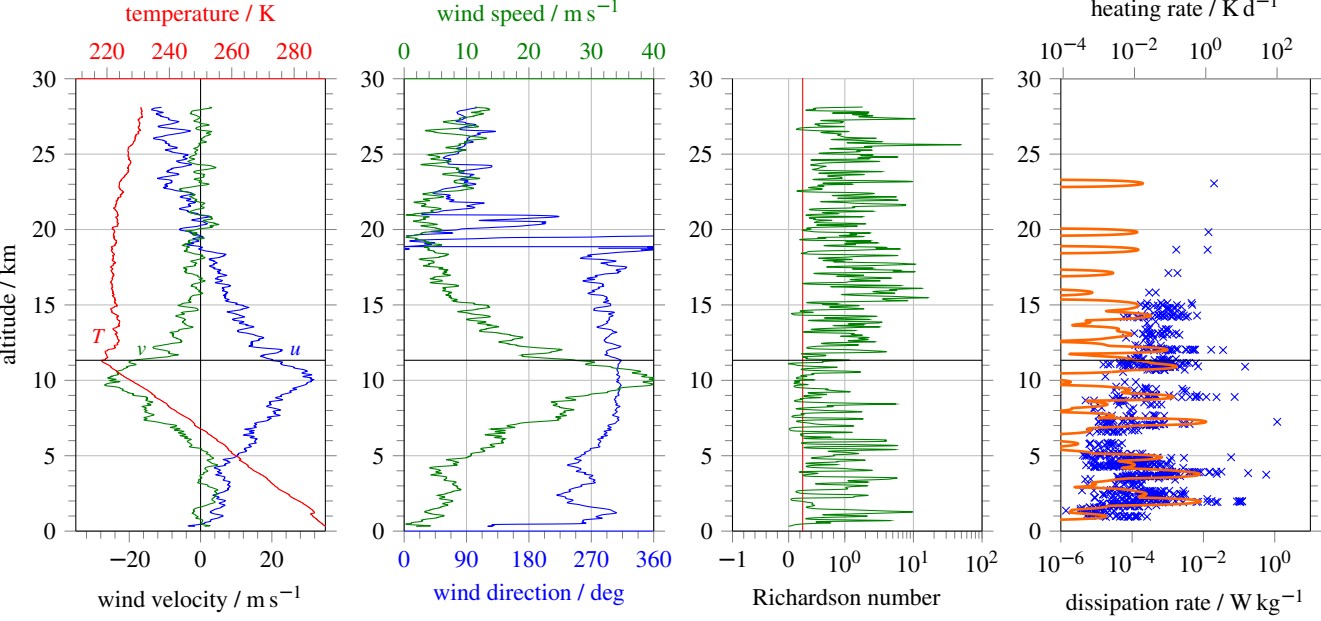

**Figure 7.** Same as Figure 1, but for the flight from Kühlungsborn at 12 Jul 2015. The dissipation profile excludes the lowermost 550 m due to disturbances from the launch procedure (dereeling of the payload suspension).





**Figure 8.** Same as Figure 2, but for WRF simulations for 11 Jul 2015, 23:00 UT