# Peer review of "Case study of wave breaking with high-resolution turbulence measurements with LITOS and WRF simulations"

_Atmospheric Chemistry and Physics, 2016_

## Referee Comment (RC1) · W. K. Hocking (Referee) · 22 Dec 2016

Referee's report on the paper

"Case study of wave-breaking with high-resolution turbulence measurements with LITOS and WRF simulations"

by W.K. Hocking

**Part A - scientific issues.**

This paper presents some high resolution measurements of turbulence using a special balloon-mounted wind anemometer. The paper concentrates on 4 campaigns, and uses WRF simulations as backup.

My first impression is that the paper is trying too hard to justify the idea of "being the first" on a number of fronts. It is not necessary for a paper to always "be the first", and a good paper can make meaningful contributions even if such status is not valid. In this case, I feel that the paper over-reaches in this area. It claims that, to the knowledge of the authors, "currently the only instrument for the direct in situ observation of turbulent wind fluctuations in the middle stratosphere is the balloon-borne instrument Leibniz Institute Turbulence Observations in the Stratosphere (LITOS)". Other statement pertaining to the uniqueness of the paper occur elsewhere in the text, where the authors discuss the Richardson number, and the essentially repeat the studies of Hines (1988), who introduced the idea of the "slant-wise instability" as far back as 1988.

The first measurements of velocity fluctuations in the stratosphere using balloon-borne instruments was due to Barat (1982), which the authors do refer to later, but fail to give it due recognition. More recently, extensive measurements (including velocity, temperature and humidity) have been presented by Cho et al., (Cho, J.Y.N., Newell,R.E., Anderson,B.E., Barrick,J.D.W., Thornhill,K.L., 2003, Characterizations of tropospheric turbulence and stability layers from aircraft observations.J.Geophys.Res.108(D20),8784. http://dx.doi.org/10.1029/ 2002JD0082820) and Cornman ((1) Cornman, L.B., Corrine, S.M., Cunning, G., 1995, Real-time estimation of atmospheric turbulence severity from in-situ aircraft measurements, J. Aircraft 32, 171–177.; and (2) Cornman, L.B., Meymaris, G., Limber, M., 2004, An update on the FAA Aviation Weather Research Program's in situ turbulence measurement and report system. Preprints.In:11th Conference on Aviation, Range, and Aerospace Meteorology, Hyannis, MA, Amer. Meteor. Soc. CD-ROM, P4.3.f.

None of these latter works are referenced. It also should be noted that Dehghan et al., ("Comparisons between multiple in-situ aircraft turbulence measurements and radar in the troposphere", J. Atmos. Solar-Terr. Phys., 118(A), 64-77, http://dx.doi.org/10.1016/j.jastp.2013.10.009, 2014) have found errors in the calibration of the papers by Cornman et al.

It is true that the procedures used in the paper under review are probably the most detailed I have seen, but they are not the only ones. The procedure used by Cornman and colleagues, for example, places accelerometers on commercial aircraft and measures turbulent fluctuations. The results are of course heavily filtered because only scales comparable to and larger than the size of the aircraft are measured,

which partly accounts for the corrections introduced by Dehghan et al (2014). However, the sheer magnitude of measurements by this technique is staggering, and vastly outweighs the measurements by LITOS - an important aspect for studies of large-scale diffusion, as will be discussed below. The works by Cho et al. are much more thorough.

Nevertheless, the fact remains that LITOS is not the only instrument used for these studies, nor is it the only instrument which measures velocity fluctuations

There are a variety of instances when the authors do not give due recognition. The work of Barat, Sidi, Wilson etc., who have spent over 30 years studying stratospheric turbulence - mainly with temperature probes - have not been mentioned in the introduction. (see the references in Osman et al., 2016, discussed below).

So these references need to be added.

However, rather than simply being critical, I would like to offer an alternative approach to the introduction. I first invite the authors to look at the introduction to Osman et al., (2016) viz. Osman, M.K., W. K. Hocking and D. W. Tarasick, "Parameterization of Large-Scale Turbulent Diffusion in the presence of both well-mixed and weakly mixed patchy layers", J. Atmos. Solar-Terr. Phys., 143-144 14-36, 2016.

I will summarize this work below. This summary can be used to place the work presented by Schneider et al. in a far more useful context. The current authors have simply justified their paper on the need for measurement of turbulence in the stratosphere. But in fact there are much larger and more important issues at play here which are of great physical significance, unrecognized by a large portion of the community but of great relevance.

The issue is the following. In 1981, Dewan (1981) studied the effects of small layers of fully developed turbulence, separated by regions of laminar flow, on large scale diffusion in the upper troposphere and stratosphere. This work was referenced several times following this (e.g. Hocking, W. K., The effects of middle atmosphere turbulence on coupling between atmospheric regions, J. Geomag. Geoelectr., 43, Suppl., 621-636, 1991; Hocking, W. K., The dynamical parameters of turbulence theory as they apply to middle atmosphere studies, Earth, Planets and Space, 51 , 525-541, 1999).

In the 1990's, Haynes and co-authors presented a series of papers following a similar theme; citations of these papers can be found in Vanneste, J., Small-scale mixing, large-scale advection, and stratospheric tracer distributions, *J. Atmos. Sci.*, *61* , 2749–2761, 2004.

The basic premise was that thin layers of turbulence, separated by essentially laminar regions of flow, are the primary form of turbulence in the stratosphere, and that in determining the large-scale diffusion coefficient (scales of 10 km and more) the effect of these isolated layers is

paramount.  Equally importantly, the rate of large scale diffusion is independent of the strength of turbulence within the layers, (as long as mixing is complete), and it is other things like the frequency of occurrence of these layers, their mean depth, and their relative fraction of occurrence, which defines the large-scale diffusion process. This was a major departure from established thinking.

However, issues remained. One such issue was a proper definition of the meaning of "well-mixed". Others were the 1-D nature of the models used, and the impact of partially mixed layers. Vanneste (2004 - see above) attempted inclusion of the impact of partially mixed layers.

Osman et al. (2016) (referenced above) expanded the previous work to  a 2D model, developed a proper definition of the meaning of "well-mixed", and went on to study the impact of both small and large layers o the gross 2-D flow. (Another important process, namely that of Stokes Diffusion, was also discussed, though I will not dwell on this here).

The importance of this work to the current paper is as follows: earlier works suggested that large-scale stratospheric diffusion rates were determined by the many thousands of small layer of turbulence, whereas the work of Osman et al. suggests that it is the small number of large layers which dominate the diffusion process. The work presented in the paper under review can help resolve this critical question of how large scale diffusion relates to small-scale turbulent layering, and in so doing will have major impact on the parameterization of large-scale 2D stratospheric models  - including WRF. I am not asking that the authors resolve the issue - but simply that they link their measurements to this critical debate.

A discussion along the lines given above in the introduction will strengthen the paper enormously and highlight this critical issue of the relation between localized turbulence and large -scale diffusion. Unfortunately, despite the incredible importance of this issue, it is not as widely understood as it should be, and this is an excellent chance to emphasize this issue. Bringing the issues to the forefront can of course also allow justification for more detailed research and hence raise the profile of the issue within granting agencies.

Further detailed discussion of these issues can be found in *Hocking et al., "Atmospheric Radar: Application and Science of MST Radars in the Earth's Mesosphere,Stratosphere, Troposphere and weakly Ionized Regions", Cambridge Press, 2016 (see the discussion around Fig. 11.23).*

**I therefore ask that the authors substantially revise their introduction, add the references cited, discuss in more detail the work of Sidi, Wilson etc., (see references in Osman et al, 2016) and demonstrate the nature of their work within the context of this important discussion.**

Four other scientific issues should be considered.

**Richardson number**; First, in several places the authors discuss the relevance of the Richardson number. They conclude that this only applies in purely horizontal flows, whereas gravity waves are 3D e.g. page 6, para 2, lines 4-5 and elsewhere. They then discuss their own resolution of the problem using Achatz (2005). However, this problem was raised and discussed by Hines in 1988 (Hines, C. O., Generation of Turbulence by Atmospheric Gravity Waves, J. Atmos. Sci., 45, 1269–1278, 1988) and a citation of his work is very much deserved in this context.

**Shedding**: The authors discuss the idea that wave breaking is not due to single waves, but due to multiple waves adding together, and also the idea that the waves do not break catastrophically, but at times simply throw off just enough energy to allow them to become stable. This is a process called "shedding", or alternatively "convective adjustment", which is very well documented in the literature. The authors refer to it as "continuous fractional wave breaking" (page 11, paragraph 3). It is very important that the authors (again) give credit to those who have gone before them. References, and extensive discussion, can be found in Hocking, W.K., A review of Mesosphere–Stratosphere–Troposphere(MST) radar developments and studies, circa 1997–2008, Journal of Atmospheric and Solar-Terrestrial Physics (2010), doi:10.1016/j.jastp.2010.12.009, section 8. An even more extended discussion can be found in *Atmospheric Radar: Application and Science of MST Radars in the Earth's Mesosphere, Stratosphere, Troposphere and weakly Ionized Regions", Cambridge Press, 2016, chapter 11, section 11.2.12*. (Note that Hocking was not the main person who proposed this method, but has summarized the many different papers on the technique- the authors are asked to use these to references simply as a starting point for their own clarification, and to properly cite those who have discussed the method in more detail).
Another paper on a similar topic, but "tuned" to the upper troposphere and stratosphere, is Fairall, C. W., A. B. White, and D. W. Thomson, A stochastic model of gravity-wave-induced clear-air turbulence, *J. Atmos. Sci.*, *48* , 1771–1790, 1991.

**Equation (1) and appendix A**: Equation (1) and appendix A take an interesting approach to determination of the energy dissipation rate (epsilon). The traditional method for determining epsilon is to determine structure functions, or spectra, and fit relevant Kolmogoroff functions to the inertial range portion. The introduction of the method given in equation (i) began in the early 1990's, when Luebken (a co-author on this paper), introduced it as an alternative procedure in order to help resolve an argument that developed in the literature when applications of the more traditional approach produced differences of almost an order of magnitude when applied by different authors. For details of the debate, the reader is referred to Hocking, W. K., The dynamical parameters of turbulence theory as they apply to middle atmosphere studies, Earth, Planets and Space, 51 , 525-541, 1999, but the argument was resolved by introduction of Luebken's approach. However, the paper just mentioned (Hocking) then showed that the inner-scale-approach and the traditional approach produced similar values as long as the correct constants were used. The problem arose because a group of

workers incorrectly used a constant pertaining to the integrated 3-D energy spectrum, whereas the measurements were made using a probe passing though the turbulence in a straight line, dictating the need for a different constant (the integrated 3-D spectrum and the 1D spectrum both have a $k^{-5/3}$ spectral form, giving rise to confusion). The constant differed by a factor of 3, so the epsilon thus deduced was in error by $3^{5/3}$ times, or about 6x. Similar problems have arisen in other areas of the literature, even quite recently. The Appendix of Hocking , EPS, 1999 (given above) and appendix A of Hocking et al., (book published by Cambridge Press and discussed above) show how to use the correct constants.

Further verification of these constants has been given by aircraft/radar comparisons in Dehghan et al., (2014) (reference given above).

Given that the issue of the correct constants is now resolved, there is no reason why the more traditional approach should not be used - and indeed it has been used for many years by a variety of authors who HAVE used the correct constants - it is simply unfortunate that from time to time papers are published by authors who apply the wrong criteria, for reasons outlined above. The use of equation (1) in the paper under review is OK, but places significant constraints on the analysis. As the author have shown, many spectra cannot be used since they do not show a "knee" in the spectrum. It would be if interest to see how the approach using $l\_0$ and the more traditional structure function/spectral approach (with correct constants) compare. The issue is important in view of the first item discussed in this review concerning the relation between small-scale and large-scale diffusion rates. The ability to measure epsilon using only the inertial part of the spectrum allows access to a larger data set, and the most important parameters could be argued to be the frequency of occurrence of turbulent layers, and their fraction of occupancy, while the actual strength of the turbulence within the layers might be less important for determination of large-scale diffusion. Hence the availability of more useable data allows a better contribution to studies of these fractions and statistics (see P 4, ln 4 - seems the study presented here is not ideal for determining percentages). If there is insufficient room to discuss it here, it at least seems good topic for future study, and I recommend it to the authors. If the author have already done such a study, they should cite it.

**Use of WRF**: The authors include substantial discussions of the wave-field inferred from the WRF model. I do find myself wondering about the validity of this approach. Are modern models really good enough to reproduce the detailed small-scale structure in real-life situations? It seems somewhat unlikely to me, but perhaps I have not kept apace with current computer developments. But treating the model output as a true representation of the wind and temperature field seems a stretch. Even if the waves are generated reasonably accurately in amplitude, variations of phase estimates can significantly impact the likelihood that they break (as per the author's on comments about "continuous fractional wave breaking" and also the

concepts presented by Fairall et al. discussed above). I feel the approach is interesting, but am concerned it is a bit premature. I am happy to see the process introduced, **but would ask for more commentary about its likely validity**.

**Part B - grammatical.**

A variety of grammatical errors occur throughout the text, which are listed below.

P 1, abstract, line 9 - Particularly --> In particular

P 1, ln 17 -  "This typically happens in the mesosphere".  What typically happen there? Are the authors talking about catastrophic wave breakdown, or shedding? As discussed in part A, wave breaking is expected in some form everywhere, either by full breakdown or multiple-wave interference effects, so I am not sure this sentence is especially useful.

 P 2, l1 - Measurements are --> Measurements have been??

Section 1 - see earlier notes in pat A - many missing references to other work.

P 2, ln 19 - use of "thereof" seems odd - suggest replacing with "comprising".

P 3, ln 1 - "booms sticking out"--> "booms protruding"

P 3, ln 8 - suggest "windows of 5m" --> "windows with depths of 5m"??

P 3, ln 25 - rejection of spectra which are "not meaningful" - seems presumptive to assume that the only acceptable spectra are ones consistent with their proposed theory, though its understandable that no useful epsilon can be acheived in such circumstances I guess.

P 4, ln 3 - suggest "conditions" --> "above conditions"

P 4, ln 4 - suggest "rigorous criteria" --> "rigorous criteria applied"

 P 4, ln 8 - "sensor has been ..." --> "sensor has been located ..."

P 4, ln 26 - reference to Hines' work on slantwise instability is needed.

P 6, ln 28 - the 30% does not seem meaningful due to the selection criteria used .

P 6, ln 2 - "with respective phase velocities"  -  with respect to what? the meaning of the sentence is quite unclear.

P 6, ln 6 - "other side" - do you mean "other hand"?

P 6, ln 19 - "It visualizes.." --> "It demonstrates..." ??

P 8, ln 1 - could change "..at 27 Mar 2014 10:10 UT." to "..on 27 Mar 2014 at 10:10 UT."

P 8 ln 7... ".. were easterly and turned northwards ..." This is a confusing mixture of directional conventions use either ".. were westward and turned northwards.." or "were easterly and turned southerly.." Similar problems exist elsewhere in the text - try to standardize directional information (meteorological directions end in "ly" and indicate the direction from which the wind comes. whereas middle-atmosphere convention more commonly ends directions in "..ward" and indicate the direction in which the wind is blowing towards. Whichever convention is used is fine, but please try to standardize.

P 8 ln 31 - "and partly even smaller..." - do you mean "and at times smaller.." ??

P 9, ln 12 - mention is made of a "layer(ed) structure" - since it is a 1D vertical profile, how ca you be sure it is really layered?

P 9, ln 26 - matybe change "yields" to "suggests" ??

P 10, paragraph 2 - while a useful summary of these data, there were only 4 flights, and these results can really only be considered as anecdotal.

P 10, last paragraph. Some of these references could be cited in the introduction.

P 11, lines 21-22 - link to pre-existing papers regarding shedding and convective adjustment rather than introducing yet another name.

=================== end ==============================

---

## Referee Comment (RC2) · Anonymous Referee #2 · 4 Jan 2017

This paper describes comparisons of four high-resolution turbulence measurements from LITOS-equipped balloons with WRF simulations, with the goal of better understanding the sources of turbulence observed in the LITOS ascents. The LITOS-derived energy dissipation rates appear to be carefully done, however the WRF simulations, while suggestive, need more attention. I would therefore recommend publication in ACP subject to major revisions that include more thorough analyses of the WRF output.

Major comments

1. Regarding the WRF simulations we are not given any evidence the simulations are correctly modeling the atmospheric environment. At a minimum, comparison of wind

and temperature profiles at the location of the balloon ascents to the LITOS profiles should be done. And there should be plenty of surface data to compare to as well. Also, what about comparisons to satellite imagery: is there any evidence of waves in the images? If so what are the wavelengths and do they agree with the WRF predicted wavelengths?

2. In a similar vein, while I agree that 2 km resolution is probably sufficient to resolve most gravity waves that may be generated either topographically or from other sources, it is not sufficient to model "wave breaking". This would require much higher resolutions. See e.g., Kim et al. MWR 2014 and Trier and Sharman, MWR 2016 for examples of the effects of model grid spacing on gravity wave resolutions.

3. Another approach might be to attempt to diagnose regions of gravity wave breaking from the LITOS or model derived soundings using standard gravity wave drag parameterizations, described e.g., in Nappo's 2002 book, and used in Kim and Chun JAMC 2011. Also looking for the presence of gravity wave critical levels in the WRF output may be useful in diagnosing regions of likely wave breaking.

4. Looking at the LITOS figures I really don't see a good correlation between epsilon and low values of Ri. This is not unexpected (e.g., Galperin et al. ASL 2007), and implies it is difficult if not impossible to assign a threshold Ri for turbulence. The authors discuss this in Section 2.1, but it should be also emphasized in the conclusions section.

Minor comments

- 1. p. 2 line 27. Do you mean a precision of 1 cm s-1?
- 2. p. 3 line 6. Do you mean "sensors" instead of "sectors"?

3. p. 3 lines 10-13. While I understand the attempt to use the Heisenberg spectrum to fit the high frequency end of the measurements, wouldn't it be simpler and less error prone to simply fit the portion of the spectrum in the inertial range to determine epsilon?

4. p. 3 line 20. How can epsilon computed from eqn (10) ever be negative when the

**ACPD**
individual terms are raised to the 4th power and are therefore even, and nu should always by positive?

5. p. 11 line 22. Could you elaborate on what is meant by "continuous fractional wave breaking"?

6. In the LITOS figures (1,3,5,7), what is heating rate on the left panel? It would be interesting to plot shear and stability as well, and this may help in assessing the character of the turbulence.

7. Appendix. The gamma function in the eqns is not defined.

---

## Referee Comment (RC3) · Anonymous Referee #3 · 17 Jan 2017

**"Case study of wave breaking with high-resolution turbulence measurements with LITOS and WRF simulations" by A. Schneider et al.**

**Submitted to Atmospheric Chemistry and Physics**

This paper describes the results of high-resolution atmospheric turbulence measurements using LITOS, which are complemented by WRF simulations of gravity waves breaking in similar atmospheric conditions. I found the analysis of the experimental part of the study carefully done and potentially useful to modelers trying to fit simulation parameters (e.g. eddy dissipation rate) to real observational data for benchmarking or validation purposes. On the other hand, I do not believe that WRF (especially in the chosen simulations set-up) is an adequate tool to carry out equivalent "high-resolution" turbulence simulations for the reasons explained below. In order to publish the paper, the authors should justify their modeling choice clearly stating the limitations of the model, and better explain the numerical set-up of the simulations. This will help the reader to better judge the analysis of the simulations and to put this part of the study into the right perspective.

Major Comments

- The main point made by the authors is that an increase in GW breaking is associated to the increase in turbulence dissipation. If the authors mean that high GW *leads* to stronger turbulence, I agree. But the I would be cautious to generalize this statement implying (as the authors say at the end of Conclusions), that turbulence in the atmosphere is generated by continuous GW activity because the latter is only *one* of the causes triggering turbulence in the atmosphere (other drivers are large-scale convention, shear instabilities, etc. which do not necessarily involve GW).

- I do not believe that WRF can provide reliable information on turbulence characteristics in the chosen simulation set-up, at least the authors didn't show substantial evidence it can. The main is reasons is of course the coarse resolution: 2km is not even close to resolve eddies in a substantial (and potentially relevant to observational data) portion of the inertial range, should turbulence develop following GW breaking. Indeed, the discussion on the simulation results rely entirely on the supposed correctness of the modeled TKE transport rather than the resolution of turbulent scales! In addition, there are no details on the TKE parameterization used in the runs so it is not clear whether such parametrizing is correctly tailored to the cases analyzed. There's a huge literature on DNS/LES modeling of turbulent stratified flows -which apply to atmospheric turbulence as wll- discussing these issues. You can refer to the review study by Brethauwer et al, JFM 2007 and to more recent works such as Kani and Waite, JFM 2014, and Paoli et al, ACP 2014 in addition to the work by Fritts and coworkers on GW breaking that you cited.

- I agree with your consideration on Richardson number and the difficulty to match the theoretical Ri=0.25 threshold for shear instability in real atmospheric situations. To support

your discussion, you may also refer to the work by Paoli et al, ACP 2014 where they used high-resolution LES (with grid sizes of order of meters) to study atmospheric turbulence at the tropopause level. They observed similar trend of Ri as a function of altitude (ex their Figs. 9-10), and discussed the impact of turbulence intensity and the sensitivity to resolution, which can also apply to the measured profiles shown in your Fig 1c, 3c etc.

- It would very much benefit to the paper showing turbulence spectra or structure functions, particularly in the inertial range, and especially for the cases of developed turbulence where an inertial range should be neatly detected.

Minor comments

- What is the reason for adding a legend of K/d in addition to W/kg in the dissipation profiles of Figures 1d, 3d, etc? In fact, I also found a little weird to label the units of dissipation rate as W/kg instead of $m^2/s^3$ or $cm^2/s^3$ which is more customary in turbulence literature.

Literature added:
- Brethouwer et al, 2007: "Scaling analysis and simulation of strongly stratified turbulent flows", Journal of Fluid Mechanics. vol. 585, pp. 343-368.
- Khani and Waite, 2014: "Buoyancy scale effects in large-eddy simulations of stratified turbulence", Journal of Fluid Mechanics. vol. 754, pp. 75-97.
- Paoli et al, 2014: "High-resolution large-eddy simulations of stably stratified flows: application to subkilometer-scale turbulence in the upper troposphere–lower stratosphere", Atmospheric Chemistry and Physics, vol. 14, pp. 5037-5055.

---

## Author Comment (AC1) · 7 Feb 2017

**Author response to Review by Wayne K. Hocking**

We thank the reviewer for his detailed review and for alerting us to literature that previously escaped our notice.

In the following, the review is quoted in italics part by part, and our response given below.

**Part A – scientific issues**

My first impression is that the paper is trying too hard to justify the idea of "being the first" on a number of fronts. It is not necessary for a paper to always "be the first", and a good paper can make meaningful contributions even if such status is not valid. In this case, I feel that the paper over-reaches in this area. It claims that, to the knowledge of the authors, "currently the only instrument for the direct in situ observation of turbulent wind fluctuations in the middle stratosphere is the balloon-borne instrument Leibniz Institute Turbulence Observations in the Stratosphere (LITOS)". Other statement pertaining to the uniqueness of the paper occur elsewhere in the text, where the authors discuss the Richardson number, and the essentially repeat the studies of Hines (1988), who introduced the idea of the "slantwise instability" as far back as 1988.

We are sorry that our phrasing was mistakable, leading to the impression of claiming to be better than we actually are. As described below, we have rewritten the introduction. The sentence under discussion regarding LITOS has been deleted. The introduction now puts our method and instrument in a better context of existing data sets from airplanes etc. Regarding the discussion about the validity of the Richardson criterion, we have now put it in a "historical" context. In fact we did not claim to be the first here, but wanted to keep the description short as this has already been described in a previous paper from our group (Haack et al, 2014). In the revised version we provide a broader discussion and proper reference of this topic inclusive the slantwise instability described by Hines.

The first measurements of velocity fluctuations in the stratosphere using balloon-borne instruments was due to Barat (1982), which the authors do refer to later, but fail to give it due recognition. More recently, extensive measurements (including velocity, temperature and humidity) have been presented by Cho et al., (Cho, J.Y.N., Newell,R.E., Anderson,B.E., Barrick,J.D.W., Thornhill,K.L., 2003, Characterizations of tropospheric turbulence and stability layers from aircraft observations.J.Geophys.Res.108(D20),8784. http://dx.doi.org/10.1029/2002JD0082820) and Cornman ((1) Cornman, L.B., Corrine, S.M., Cunning, G., 1995, Real-time estimation of atmospheric turbulence severity from in-situ aircraft measurements, J. Aircraft 32, 171–177.; and (2) Cornman, L.B., Meymaris, G., Limber, M., 2004, An update on the FAA Aviation Weather Research Program's in situ turbulence measurement and report system. Preprints.In:11th Conference on Aviation, Range, and Aerospace Meteorology, Hyannis, MA, Amer. Meteor. Soc. CD-ROM, P4.3.f.

None of these latter works are referenced. It also should be noted that Dehghan et al., ("Comparisons between multiple in-situ aircraft turbulence measurements and radar in the troposphere", J. Atmos. Solar-Terr. Phys., 118(A), 64-77, http://dx.doi.org/10.1016/j.jastp.2013.10.009, 2014) have found errors in the calibration of the papers by Cornman et al.

It is true that the procedures used in the paper under review are probably the most detailed I have seen, but they are not the only ones. The procedure used by Cornman and colleagues, for example, places accelerometers on commercial aircraft and measures turbulent fluctuations. The results are of course heavily filtered because only scales comparable to and larger than the size of the aircraft are measured, which partly accounts for the corrections introduced by Dehghan et al (2014). However, the sheer magnitude of measurements by this technique is staggering, and vastly outweighs the measurements by LITOS - an important aspect for studies of large-scale diffusion, as will be discussed below. The works by Cho et al. are much more thorough.

Nevertheless, the fact remains that LITOS is not the only instrument used for these studies, nor is it the only instrument which measures velocity fluctuations.

As mentioned above, our phrasing seems to have been mistakable, thus we have revised it. However, we did not claim that LITOS is the only instrument measuring velocities in general, but that LITOS is currently the only instrument for the in-situ measurement of wind fluctuations *in the middle stratosphere*, i. e. above airplane flight altitudes.

СЗ

Since Barat's (1982) instrument seems to be no longer in operation, and all other in situ instruments for small-scale wind measurement known to us cannot measure in the middle stratosphere, we still think our statement is correct.

We agree that the database used by Cornman et al. (1995) is much larger, but it is from commercial aircraft flying in the upper troposphere or lowermost stratosphere, not the middle stratosphere. Similarly, Cho et al. (2003) used data from aircraft with a ceiling of 8 km, i. e. tropospheric heights. In the revised introduction, we have cited Cornman et al. (1995) to point out the contrast in available data for the different heights.

There are a variety of instances when the authors do not give due recognition. The work of Barat, Sidi, Wilson etc., who have spent over 30 years studying stratospheric turbulence - mainly with temperature probes - have not been mentioned in the introduction. (see the references in Osman et al., 2016, discussed below).

So these references need to be added.

We admit that the introduction was overly short in some aspects. We have rephrased it and in this context have added some references, especially technical papers. Besides, we want to focus on wave saturation and breaking, thus we have limited the cited scientific works to those related to these topics.

However, rather than simply being critical, I would like to offer an alternative approach to the introduction. I first invite the authors to look at the introduction to Osman et al., (2016) viz. Osman, M.K., W. K. Hocking and D. W. Tarasick, "Parameterization of Large-Scale Turbulent Diffusion in the presence of both well-mixed and weakly mixed patchy layers", J. Atmos. Solar-Terr. Phys., 143-144 14-36, 2016.

[...]

The importance of this work to the current paper is as follows: earlier works suggested that large-scale stratospheric diffusion rates were determined by the many thousands of small layer of turbulence, whereas the work of Osman et al. suggests that it is the

small number of large layers which dominate the diffusion process. The work presented in the paper under review can help resolve this critical question of how large scale diffusion relates to small-scale turbulent layering, and in so doing will have major impact on the parameterization of large-scale 2D stratospheric models - including WRF. I am not asking that the authors resolve the issue - but simply that they link their measurements to this critical debate.

A discussion along the lines given above in the introduction will strengthen the paper enormously and highlight this critical issue of the relation between localized turbulence and large-scale diffusion. Unfortunately, despite the incredible importance of this issue, it is not as widely understood as it should be, and this is an excellent chance to emphasize this issue. Bringing the issues to the forefront can of course also allow justification for more detailed research and hence raise the profile of the issue within granting agencies.

We agree that the issues of intermittency and turbulent mixing are important. LITOS data are very suitable for such a study. Yet this is outside the scope of this paper, which is about wave breaking and turbulence. We want to address these issues in future work.

[...]

I therefore ask that the authors substantially revise their introduction, add the references cited, discuss in more detail the work of Sidi, Wilson etc., (see references in Osman et al, 2016) and demonstrate the nature of their work within the context of this important discussion.

We have revised the introduction, setting our measurements in a better historical context. Furthermore, we have strengthened the description of the geophysical scope.

**Richardson number**; First, in several places the authors discuss the relevance of the Richardson number. They conclude that this only applies in purely horizontal flows,

whereas gravity waves are 3D e.g. page 6, para 2, lines 4-5 and elsewhere. They then discuss their own resolution of the problem using Achatz (2005). However, this problem was raised and discussed by Hines in 1988 (Hines, C. O., Generation of Turbulence by Atmospheric Gravity Waves, J. Atmos. Sci., 45, 1269–1278, 1988) and a citation of his work is very much deserved in this context.

We thank the referee for pointing us to Hines' (1988) work which has inspired many later works. We have added a few sentences mentioning his ideas:

"Already Hines (1988) discussed slantwise static instabilities created by gravity waves. He developed a wave period criterion for turbulence by comparing the e-folding time of the (slantwise) instability with the period of the wave. Turbulence is more likely to occur for slantwise static instability than for vertical static instability."

Shedding: The authors discuss the idea that wave breaking is not due to single waves, but due to multiple waves adding together, and also the idea that the waves do not break catastrophically, but at times simply throw off just enough energy to allow them to become stable. This is a process called "shedding", or alternatively "convective adjustment", which is very well documented in the literature. The authors refer to it as "continuous fractional wave breaking" (page 11, paragraph 3). It is very important that the authors (again) give credit to those who have gone before them. References, and extensive discussion, can be found in Hocking, W.K., A review of Mesosphere–Stratosphere–Troposphere(MST) radar developments and studies, circa 1997–2008, Journal of Atmospheric and Solar-Terrestrial Physics (2010), doi:10.1016/j.jastp.2010.12.009, section 8. An even more extended discussion can be found in "Atmospheric Radar: Application and Science of MST Radars in the Earth's Mesosphere, Stratosphere, Troposphere and weakly Ionized Regions", Cambridge Press, 2016, chapter 11, section 11.2.12. (Note that Hocking was not the main person who proposed this method, but has summarized the many different papers on the technique- the authors are asked to use these to references simply as a starting point for their own clarification, and to properly cite those who have discussed the method in more detail). Another paper on a similar topic, but "tuned" to the upper troposphere and stratosphere, is Fairall, C. W., A. B. White, and D. W. Thomson, A stochastic model of gravity-wave-induced clear-air turbulence, J. Atmos. Sci., 48, 1771–1790, 1991.

We thank the reviewer for pointing us to the correct terminology. A literature search starting from the articles given above has yielded that "saturation" seems to be the most commonly used term for the phenomenon. Thus we have changed our manuscript accordingly.

Equation (1) and appendix A: Equation (1) and appendix A take an interesting approach to determination of the energy dissipation rate (epsilon). The traditional method for determining epsilon is to determine structure functions, or spectra, and fit relevant Kolmogoroff functions to the inertial range portion. The introduction of the method given in equation (i) began in the early 1990's, when Luebken (a co-author on this paper), introduced it as an alternative procedure in order to help resolve an argument that developed in the literature when applications of the more traditional approach produced differences of almost an order of magnitude when applied by different authors. For details of the debate, the reader is referred to Hocking, W. K., The dynamical parameters of turbulence theory as they apply to middle atmosphere studies, Earth, Planets and Space, 51, 525-541, 1999, but the argument was resolved by introduction of Luebken's approach. However, the paper just mentioned (Hocking) then showed that the innerscale-approach and the traditional approach produced similar values as long as the correct constants were used. The problem arose because a group of workers incorrectly used a constant pertaining to the integrated 3-D energy spectrum, whereas the measurements were made using a probe passing though the turbulence in a straight line, dictating the need for a different constant (the integrated 3-D spectrum and the 1D spectrum both have a  $k^{-5/3}$  spectral form, giving rise to confusion). The constant differed by a factor of 3, so the epsilon thus deduced was in error by  $3^{5/3}$  times, or about 6x. Similar problems have arisen in other areas of the literature, even quite recently. The Appendix of Hocking, EPS, 1999 (given above) and appendix A of Hocking

et al., (book published by Cambridge Press and discussed above) show how to use the correct constants. Further verification of these constants has been given by aircraft/radar comparisons in Dehghan et al., (2014) (reference given above). Given that the issue of the correct constants is now resolved, there is no reason why the more traditional approach should not be used - and indeed it has been used for many years by a variety of authors who HAVE used the correct constants - it is simply unfortunate that from time to time papers are published by authors who apply the wrong criteria, for reasons outlined above. The use of equation (1) in the paper under review is OK, but places significant constraints on the analysis. As the author have shown, many spectra cannot be used since they do not show a "knee" in the spectrum. It would be if interest to see how the approach using  $l_0$  and the more traditional structure function/spectral approach (with correct constants) compare. The issue is important in view of the first item discussed in this review concerning the relation between small-scale and largescale diffusion rates. The ability to measure epsilon using only the inertial part of the spectrum allows access to a larger data set, and the most important parameters could be argued to be the frequency of occurrence of turbulent layers, and their fraction of occupancy, while the actual strength of the turbulence within the layers might be less important for determination of large-scale diffusion. Hence the availability of more useable data allows a better contribution to studies of these fractions and statistics (see P 4, In 4 - seems the study presented here is not ideal for determining percentages). If there is insufficient room to discuss it here, it at least seems good topic for future study, and I recommend it to the authors. If the author have already done such a study, they should cite it.

For our measurement the "traditional" method to fit the inertial range of the spectrum is not possible, because that method crucially depends on the absolute value of the periodogram, which is not available due to missing calibration. A calibration to infer wind velocities from the anemometer voltage of the constant temperature anemometer would be difficult because it has to be performed in a laboratory for known velocities under the same ambient conditions for pressure and temperature as the measurement.

Conditions of a balloon flight, where pressure varies within several orders of magnitude and temperature changes by  ${\sim}80\,\text{K}$ , are very difficult to simulate in a wind tunnel. We do not know a facility where such a calibration would be possible.

We agree that a comparison of dissipation rates from both retrievals, i. e. the traditional inertial range method and Lübken's method, would be very interesting. We have planned to do such a comparison for a measurement on the ground where the calibration problem can be solved with relative ease.

**Use of WRF**: The authors include substantial discussions of the wave-field inferred from the WRF model. I do find myself wondering about the validity of this approach. Are modern models really good enough to reproduce the detailed small-scale structure in real-life situations? It seems somewhat unlikely to me, but perhaps I have not kept apace with current computer developments. But treating the model output as a true representation of the wind and temperature field seems a stretch. Even if the waves are generated reasonably accurately in amplitude, variations of phase estimates can significantly impact the likelihood that they break (as per the author's on comments about "continuous fractional wave breaking" and also the concepts presented by Fairall et al. discussed above). I feel the approach is interesting, but am concerned it is a bit premature. I am happy to see the process introduced, **but would ask for more commentary about its likely validity.**

We agree that WRF is an idealised representation and does not reproduce reality in a perfect way. In our paper it is used to get an overview of the respective meteorological situation during the LITOS flights and to demonstrate that gravity waves occured in the vicinity of the flight tracks. Our interpretation of the model results is not based the on small-scale structures, but on the general dynamics. Obviously, in some cases (e.g. the BEXUS 12 flight) small-scale dynamics in WRF is at least qualitatively correct and produces turbulent layers that were also found prominent in our observations. There was a good agreement between observed increase in dissipation rates and intensified TKE in the model. On the other hand, we intentionally do not investigate and interpret

the many cases where LITOS observes turbulence and WRF not. All our statements derived from WRF are based on well-resolved events.

We agree that a general validation of model results was missing and have added plots of winds and temperatures from WRF interpolated along the trajectory to the plots of the radiosonde measurements. These compare very well, the only difference is that the radiosonde data contain signatures from small-scale gravity waves which WRF cannot resolve.

Part B - grammatical

We thank the referee for the detailed grammatical corrections. We appreciate the effort.

P 1, abstract, line 9 - Particularly  $\rightarrow$  In particular

Changed.

*P* 1, In 17 - "This typically happens in the mesosphere". What typically happen there? Are the authors talking about catastrophic wave breakdown, or shedding? As discussed in part A, wave breaking is expected in some form everywhere, either by full breakdown or multiple-wave interference effects, so I am not sure this sentence is especially useful.

We have removed this sentence in our revised version of the introduction. Instead, we have written: "This mechanism has been suggested by Hodges (1967) to explain turbulence in the mesosphere."

*P 2, I1 - Measurements are*  $\rightarrow$  *Measurements have been??*

This sentence has been removed in the revision process.

P 2, In 19 - use of "thereof" seems odd - suggest replacing with "comprising".

Changed.

P 3, In 1 - "booms sticking out"  $\rightarrow$  "booms protruding"

Changed.

P 3, In 8 - suggest "windows of 5m"  $\rightarrow$  "windows with depths of 5m"??

Changed.

*P* 3, In 25 - rejection of spectra which are "not meaningful" - seems presumptive to assume that the only acceptable spectra are ones consistent with their proposed theory, though its understandable that no useful epsilon can be achieved in such circumstances I guess.

We state that if the bend in the spectrum is not resolved, the *fit* is not meaningful (not the spectrum). That means no  $\varepsilon$  can be retrieved using Heisenberg's model. We have added a phrase to clarify that:

"This means that the bend in the spectrum is not within the fit range and thus the fit is not meaningful, allowing no useful retrieval of  $\varepsilon$ ."

Generally, we only consider spectra that follow the turbulence model, which may exclude turbulence that is not fully developed. The criteria sort out cases where  $\varepsilon$  cannot be retrieved. In our manuscript we have added two sentences discussing this issue:

"Requiring the spectrum to follow Heisenberg's turbulence model may exclude turbulence that is not fully developed. However, it is not feasible to retrieve  $\varepsilon$  in cases where the periodogram does not follow the turbulence model."

P 4, In 3 - suggest "conditions"  $\rightarrow$  "above conditions"

Changed.

P 4, In 4 - suggest "rigorous criteria" → "rigorous criteria applied"

Changed.

P 4, In 8 - "sensor has been ..."  $\rightarrow$  "sensor has been located ..."

**Changed.**

P 4, In 26 - reference to Hines' work on slantwise instability is needed.

Done, see response above under Part A.

P 5, In 28 - the 30% does not seem meaningful due to the selection criteria used.

We have rephrased the sentence to make clear the 30 % is according to the criteria presented in Section 2.1:

"Overall,  ${\sim}30\,\%$  of the atmosphere was turbulent according to the criteria presented in Section 2.1."

*P* 6, In 2 - "with respective phase velocities" - with respect to what? the meaning of the sentence is quite unclear.

We have rephrased the sentence as "caused filtering of gravity waves with phase velocities equal to the background winds (if present)."

P 6, In 6 - "other side" - do you mean "other hand"?

Yes, changed.

P 6, In 19 - "It visualizes.."  $\rightarrow$  "It demonstrates..." ??

Changed.

P 8, In 1 - could change "...at 27 Mar 2014 10:10 UT." to "...on 27 Mar 2014 at 10:10 UT."

Changed, also for the other flights.

*P* 8 In 7... ".. were easterly and turned northwards ..." This is a confusing mixture of directional conventions use either ".. were westward and turned northwards.." or "were easterly and turned southerly." Similar problems exist elsewhere in the text - try to standardize directional information (meteorological directions end in "ly" and indicate the direction from which the wind comes. whereas middle-atmosphere convention

more commonly ends directions in "..ward" and indicate the direction in which the wind is blowing towards. Whichever convention is used is fine, but please try to standardize.

This was an error. Northwards has been corrected to northerly.

P 8 In 31 - "and partly even smaller..." - do you mean "and at times smaller.." ??

Yes, changed.

*P* 9, In 12 - mention is made of a "layer(ed) structure" - since it is a 1D vertical profile, how ca you be sure it is really layered?

We cannot be sure about the horizontal extension of the layers. Our use of the term "layer" stemmed from the general belief that turbulence occurs in pancake-shaped layers of a few 10 m vertical and several km horizontal extent, which is supported by radar and aircraft measurements. To avoid misunderstandings, we have changed our phrasing to "patchy structure".

P 9, In 26 - maybe change "yields" to "suggests" ?

Changed.

*P* 10, paragraph 2 - while a useful summary of these data, there were only 4 flights, and these results can really only be considered as anecdotal.

A phrase was added to clarify that averages over altitude for single flights are meant.

P 10, last paragraph. Some of these references could be cited in the introduction.

All of these references except Wilson et al. (2014) are already cited in the introduction.

*P* 11, lines 21-22 - link to pre-existing papers regarding shedding and convective adjustment rather than introducing yet another name.

As mentioned above, we have changed the terminology to wave saturation, and have cited papers discussing it.

---

## Author Comment (AC2) · 7 Feb 2017

**Author response to Anonymous Referee #2**

Major comments

1. Regarding the WRF simulations we are not given any evidence the simulations are correctly modeling the atmospheric environment. At a minimum, comparison of wind and temperature profiles at the location of the balloon ascents to the LITOS profiles should be done. And there should be plenty of surface data to compare to as well. Also, what about comparisons to satellite imagery: is there

any evidence of waves in the images? If so what are the wavelengths and do they agree with the WRF predicted wavelengths?

We agree that a validation of our WRF simulations was missing in the manuscript. We have plotted WRF data for winds and temperatures along the LITOS ascents, which shows that WRF captures the atmospheric structures well. We have also cited the paper Ehard et al. 2016, which shows a combination of WRF simulations with lidar and radiosonde data over northern Scandinavia with nearly the same model set-up as in our paper and demonstrates the ability of WRF to properly simulate GW events. Our interpretation of the WRF data is not based on specific wave parameters. For instance, we do not extract wavelengths from WRF and the exact wavelengths are not important for our reasoning. Thus, a detailed comparison of gravity wave parameters in WRF with observations is not necessary and outside the scope of this paper.

2. In a similar vein, while I agree that 2 km resolution is probably sufficient to resolve most gravity waves that may be generated either topographically or from other sources, it is not sufficient to model "wave breaking". This would require much higher resolutions. See e.g., Kim et al. MWR 2014 and Trier and Sharman, MWR 2016 for examples of the effects of model grid spacing on gravity wave resolutions.

We agree that our WRF simulations cannot simulate GW breaking of small-scale GWs with horizontal wavelengths smaller than about 10 km. Wave breaking can, however, also occur for larger-scale GWs, which are explicitly resolved by the model. Ehard et al. 2016 show regions of wave breaking at altitudes between 25 km to 30 km by means of convective overturning and reduced Richardson numbers, which was simulated by WRF with grid distances of 2 km. In our paper we use the TKE output from the model, which is provided by the boundary layer scheme and shows regions of intensified turbulent mixing in the atmosphere. For some flights (e.g., BEXUS 12, Fig. 2) these regions agree well with regions of
increased dissipation rates from LITOS. Apart from that, we do not discuss turbulence that is not resolved in WRF, but observed by LITOS, since of course WRF cannot resolve all details that LITOS can measure. We added some sentences and citations about this issue in Section 2.2. Moreover, we have added a sentence at the end of Section 3.1 stating that we intentionally do not examine turbulence unresolved in WRF.

3. Another approach might be to attempt to diagnose regions of gravity wave breaking from the LITOS or model derived soundings using standard gravity wave drag parameterizations, described e.g., in Nappo's 2002 book, and used in Kim and Chun JAMC 2011. Also looking for the presence of gravity wave critical levels in the WRF output may be useful in diagnosing regions of likely wave breaking.

Kim and Chun 2011 diagnosed turbulence sources for a large dataset by looking at lightning data for convective generation, reanalysis data for shear-induced turbulence from jet streams, and a digital elevation model for mountain waves. While this probably works for statistical statements as done by Kim and Chun 2011, we think it will not work for individual cases.

4. Looking at the LITOS figures I really don't see a good correlation between epsilon and low values of Ri. This is not unexpected (e.g., Galperin et al. ASL 2007), and implies it is difficult if not impossible to assign a threshold Ri for turbulence. The authors discuss this in Section 2.1, but it should be also emphasized in the conclusions section.

We have added a respective paragraph in the conclusions:

"Turbulence has been observed for Richardson numbers below as well as above the critical number of 1/4, partly even for values larger than 100. Such a violation of the classical theory by Miles (1961) and Howard (1961) has already been described by several researchers, e.g. Achatz (2005); Galperin (2007); Haack (2014). Hines (1988) recognised the limitation of considering only vertical instaInteractive comment

bility (as done when using the Richardson number) and proposed a concept of slantwise instabilities as created by gravity waves. He showed that turbulence is more likely to develop via slanted instability. Thus turbulence for Ri > 1/4 is comprehensible."

Minor comments

1. p. 2 line 27. Do you mean a precision of 1 cm s-1?

We mean a precision of a few  $cm s^{-1}$ . The sentence was changed accordingly.

- p. 3 line 6. Do you mean "sensors" instead of "sectors"? Yes, changed.
- 3. p. 3 lines 10-13. While I understand the attempt to use the Heisenberg spectrum to fit the high frequency end of the measurements, wouldn't it be simpler and less error prone to simply fit the portion of the spectrum in the inertial range to determine epsilon?

Deriving  $\varepsilon$  from fitting the inertial range of the spectrum is not possible for our measurement. For this method, the dissipation rate crucially depends on the absolute value of the periodogram, which is unknown due to missing calibration. A calibration to infer wind velocities from the anemometer voltage would be difficult because it has to be performed in a laboratory for known velocities under the same ambient conditions for pressure and temperature as the measurement. Conditions of a balloon flight, where pressure varies within several orders of magnitude and temperature changes by ~80 K, are very difficult to simulate in a wind tunnel. We do not know a facility where such a calibration would be possible.

4. p. 3 line 20. How can epsilon computed from eqn (10) ever be negative when the individual terms are raised to the 4th power and are therefore even, and nu should always by positive?

ACPD
We thank the reviewer for pointing at this issue. The condition stems from an earlier version of our retrieval a few years back when  $\varepsilon$  was fitted instead of  $l_0$ , which numerically allowed negative values of  $\varepsilon$  to be returned by the fitting procedure (equation (1) had been incorporated in the fit function). With the changed fit parameter this is no more possible. Thus the condition can now safely be removed. In the manuscript, this item has been deleted.

5. p. 11 line 22. Could you elaborate on what is meant by "continuous fractional wave breaking"?

We have changed the term to "wave saturation".

6. In the LITOS figures (1,3,5,7), what is heating rate on the left panel? It would be interesting to plot shear and stability as well, and this may help in assessing the character of the turbulence.

On the right panel, the top axis gives the heating rate due to turbulent dissipation,  $dT/dt = \varepsilon/c_p$ . A sentence was added in the figure caption to explain that: "The top axis gives the heating rate due to turbulent dissipation,  $dT/dt = \varepsilon/c_p$ ."

Since the Richardson number does not correspond well to turbulence (cf. major comment 4), splitting  $R_i$  in wind shear and buoyancy frequency seems not to provide useful information.

7. Appendix. The gamma function in the eqns is not defined.

A phrase defining it was added: "where [...]  $\Gamma(z) := \int_0^\infty t^{z-1} e^{-t} dt$  is the Gamma function, ..."

**ACPD**

---

## Author Comment (AC3) · 7 Feb 2017

**Author response to Anonymous Referee #3**

Major comments

• The main point made by the authors is that an increase in GW breaking is associated to the increase in turbulence dissipation. If the authors mean that high GW leads to stronger turbulence, I agree. But the I would be cautious to generalize this statement implying (as the authors say at the end of Conclusions), that turbulence in the atmosphere is generated by continuous GW activity because

the latter is only one of the causes triggering turbulence in the atmosphere (other drivers are large-scale convention, shear instabilities, etc. which do not necessarily involve GW).

We mean that high GW leads to stronger turbulence. This is seen in our measurements by large dissipation for large GW amplitudes and low dissipation for small GW amplitudes. Of course, other sources could contribute to turbulence as well when present. At the end of our conclusions, we have removed the word "generally" which may have been mistakable, i. e. the sentence now reads: "Altogether, observed dissipation is weaker during lower wave activity (as seen in WRF), and larger where larger wave amplitudes are seen."

I do not believe that WRF can provide reliable information on turbulence characteristics in the chosen simulation set-up, at least the authors didn't show substantial evidence it can. The main is reasons is of course the coarse resolution: 2 km is not even close to resolve eddies in a substantial (and potentially relevant to observational data) portion of the inertial range, should turbulence develop following GW breaking. Indeed, the discussion on the simulation results rely entirely on the supposed correctness of the modeled TKE transport rather than the resolution of turbulent scales! In addition, there are no details on the TKE parameterization used in the runs so it is not clear whether such parametrizing is correctly tailored to the cases analyzed. There's a huge literature on DNS/LES modeling of turbulent stratified flows -which apply to atmospheric turbulence as wll- discussing these issues. You can refer to the review study by Brethauwer et al, JFM 2007 and to more recent works such as Kani and Waite, JFM 2014, and Paoli et al, ACP 2014 in addition to the work by Fritts and coworkers on GW breaking that you cited.

We agree that our WRF simulations cannot simulate GW breaking of small-scale GWs with horizontal wavelengths smaller than about 10 km. Wave breaking can, however, also occur for larger-scale GWs, which are explicitly resolved by the

ACPD
model. Ehard (2016) show regions of wave breaking at altitudes between 25 km to 30 km by means of convective overturning and reduced Richardson numbers, which was simulated by WRF with grid distances of 2 km. In our paper we use WRF to get an overview of the meteorological situation and to detect regions along the balloon ascent, where increased subgrid-scale turbulent diffusion (increased TKE) was simulated by means of the boundary layer scheme. This scheme is described in Nakanishi and Niino (2009). We added some additional sentences and citations about this issue in Section 2.2.

 I agree with your consideration on Richardson number and the difficulty to match the theoretical Ri=0.25 threshold for shear instability in real atmospheric situations. To support your discussion, you may also refer to the work by Paoli et al, ACP 2014 where they used high- resolution LES (with grid sizes of order of meters) to study atmospheric turbulence at the tropopause level. They observed similar trend of Ri as a function of altitude (ex their Figs. 9- 10), and discussed the impact of turbulence intensity and the sensitivity to resolution, which can also apply to the measured profiles shown in your Fig 1c, 3c etc.

We agree that the vertical resolution has an impact on the Richardson number. Usually, a larger vertical resolution (i.e. smaller scales resolved) yields locally smaller Ri because for lower resolution Ri is potentially averaged over regions with low and high Ri. This has already been examined, e.g., by Balsley et al. (2008) and Haack et al. (2014). We have added a few sentences in our manuscript discussing this issue:

"It should be kept in mind that the Richardson number depends on the scale on which it is computed (e.g. Balsley et al. 2008; Haack et al. 2014). A higher resolution (i. e. computing Ri on smaller scales) may result in locally smaller Rinumbers, because the computation on large scales yields a kind of average. Similarly, Paoli et al. (2014) found in LES simulations larger Richardson numbers for smaller model resolutions (i. e. larger scales). Here, due to measurement noise
a smoothing over 150 m has been applied before computing Ri, determining the resolution. However, this issue cannot explain the whole discrepancy. Haack et al. (2014) examined the impact of the scale on which Ri is computed on the relation between small Richardson numbers and turbulence. They found many turbulent patches for Ri > 1 even even when computing Ri on a scale of 10 m."

• It would very much benefit to the paper showing turbulence spectra or structure functions, particularly in the inertial range, and especially for the cases of developed turbulence where an inertial range should be neatly detected.

Examples of anemometer voltages and corresponding spectra for the LITOS retrievals are shown in previous papers, e.g. Theuerkauf et al. (2011); Haack et al. (2014); Schneider et al. (2015). Since in principle we use the same retrieval, these are not shown again.

**Minor comments**

• What is the reason for adding a legend of K/d in addition to W/kg in the dissipation profiles of Figures 1d, 3d, etc? In fact, I also found a little weird to label the units of dissipation rate as W/kg instead of m2/s3 or cm2/s3 which is more customary in turbulence literature.

In some communities it is usual to give dissipation rates as heating rates in K/d, which are connected via  $dT/dt = \varepsilon/c_p$ ; thus we have added this scale as second axis. W/kg is the same as m2/s3.

---

## Author Response (AR2)

**Author response on* "Case study of wave breaking with high-resolution turbulence measurements with LITOS and WRF simulations" *by* A. Schneider et al.**

We thank the referees for their positive feedback. Below we cite the remaining minor comments together with our answers.

Additionally, we have discovered very recently that some data may be affected by instrumental effects. We have investigated this issue and deleted all data which could perhaps be affected. We choose a conservative approach and in case of doubts leave out that part of the flight. However, the main conclusion of the paper did not change, and there are still sufficient data left to support it. Please note that a substantial part of the paper, namely the WRF simulations, is not affected by the changes made. The revised manuscript with all changes marked can be found at the end of this document.

**Review by Wayne K. Hocking**

*I am OK with most of the changes, but I still believe that a reference to the diffusive effects of turbulence in the stratospheric is warranted. I suggest that at line 9 on page 2, "Regarding turbulence measurements", the following be inserted:*

*Regarding turbulence measurements,\*\*\* there are two aspects of importance: first, its energy dissipation, and secondly its diffusive properties. We will concentrate on the former: large-scale diffusion in the stratosphere is a complex process due to the intermittent nature of the turbulence there, as summarized in some detail by Osman et al., (2016), among others.\*\*\* A relatively extensive dataset exists for the troposphere and tropopause region .....*

We have inserted these sentences as suggested.

**Anonymous Referee #2**

This referee did not bring up any issues.

**Anonymous Referee #3**

*English grammar errors need attention either from the authors or from the copy editor.*

Some errors have been corrected in the revision, any potential remaining grammar errors will be addressed by the copy editor.

[revised manuscript text omitted]

~~Same as  1, but for the BEXUS 8 flight (10 Oct 2009)  **??** presents the observations. The temperature structure from the radiosonde data shows a tropopause at 8.1, i.e.considerably lower than for BEXUS 12, and only small local sections with increasing temperature above. Winds came from north western directions below ~20and from south west above. No zonal wind reversal as for BEXUS 12 was present.~~

~~Energy dissipation rates are plotted in the right panel of  **??**. Again $\epsilon$ is intermittent. In contrast to BEXUS 12, no pronounced maximum in dissipation is visible. This is consistent with the absence of a wind reversal or large wind shear. Richardson numbers are variable; mostly values are much larger than the critical number 1/4 in the entire troposphere and stratosphere, only some small layers with $Ri < 1/4$ are present. There is no extended region with $Ri < 1/4$ as for BEXUS 12 near 10altitude. Average dissipation rates are $2.0\,\mathrm{mW\,kg^{-1}}$ in the troposphere, and $5.5\,\mathrm{mW\,kg^{-1}}$ in the stratosphere (not taking into account the tropopause region 1above and below the tropopause).~~

~~Same as  2, but for WRF simulations for 10 Oct 2009, 9:00 UT and showing the trajectory of the BEXUS 8 flight. Please note that for the TKE the colourbar is scaled differently than in  2. Model simulations for the BEXUS 8 flight interpolated to the flight trajectory are plotted in the left panel of  **??**. Again, the agreement with the observations is excellent. A snapshot for the middle of the ascent is presented in  **??**. 
[revised manuscript text omitted]

---

## Author Response (AR4)

**Author response on* "Case study of wave breaking with high-resolution turbulence measurements with LITOS and WRF simulations" *by* A. Schneider et al.**

**Comments by the editor:**

*I am pleased to accept this version of the paper for publication in ACP.*

We thank the editor for his positive decision.

*Before providing a final version for publication please can you consider the following point and perhaps make an appropriate technical correction.*

*I felt the following text on p4 was somewhat confusion.*

*"This problem does not occur for the BEXUS flights, where the sensors were placed further away from the supporting lines. For all other altitude bins the average of both sensors is taken.*

*The BEXUS flight had a comparatively small distance between balloon and payload of 50 m. Thus, during considerable times the payload flew through the wake of the balloon. Therefore, only limited altitude sections with large wind shears are considered for this flight."*

*You seem to be saying first that the BEXUS flights do not have a problem and then that they do have a problem. But perhaps the problem mentioned in the second sentence is different from the problem mentioned in the first sentence. It it is then the text would be clearer if you said that explicitly.*

The first statement is about wake from the *ropes*, the second one about wake from the *balloon*. We have clarified that as follows:

[revised manuscript text omitted]